# PPDPF is not a key regulator of human pancreas development

Markus Breunig[1]*, Meike Hohwieler[1], Jasmin Haderspeck[2], Felix von Zweydorf[3], Natalie Hauff[1], Lino-Pascal Pasquini[1], Christoph Wiegreffe[4], Eleni Zimmer[1], Medhanie A. Mulaw[5], Cécile Julier[6], Eric Simon[7,8], Christian Johannes Gloeckner[3,9], Stefan Liebau[2‡], Alexander Kleger[1,10‡**]

**1** Institute of Molecular Oncology and Stem Cell Biology (IMOS), Ulm University Hospital, Ulm, Germany, **2** Institute of Neuroanatomy & Developmental Biology (INDB), Eberhard Karls University Tübingen, Tübingen, Germany, **3** DZNE-German Center for Neurodegenerative Diseases, Tübingen, Germany, **4** Institute of Molecular and Cellular Anatomy, Ulm University, Ulm, Germany **5** Central Unit Single Cell Sequencing, Medical Faculty, Ulm University, Ulm, Germany, **6** Institut Cochin, Inserm U1016-CNRS UMR8104-Université Paris Descartes, Paris, France, **7** Cardio Metabolic Diseases Research, Boehringer Ingelheim Pharma GmbH & Co KG, Biberach, Germany, **8** Computational Biology & Genomics, Boehringer Ingelheim Pharma GmbH & Co KG, Biberach, Germany, **9** Institute for Ophthalmic Research, Eberhard Karls University Tübingen, Tübingen, Germany, **10** Division of Interdisciplinary Pancreatology, Department of Internal Medicine I, Ulm University Hospital, Ulm, Germany

‡ These authors jointly supervised the work.
* markus.breunig@uni-ulm.de (MB); alexander.kleger@uni-ulm.de (AK)

## Abstract

Given their capability to differentiate into each cell type of the human body, human pluripotent stem cells (hPSCs) provide a unique platform for developmental studies. In the current study, we employed this cell system to understand the role of pancreatic progenitor differentiation and proliferation factor (PPDPF), a protein that has been little explored so far. While the zebrafish orthologue *exdpf* is essential for exocrine pancreas specification, its importance for mammalian and human development has not been studied yet. We implemented a four times CRISPR/Cas9 nicking approach to knockout *PPDPF* in human embryonic stem cells (hESCs) and differentiated PPDPF$^{KO/KO}$ and PPDPF$^{WT/WT}$ cells towards the pancreatic lineage. In contrast to data obtained from zebrafish, a very modest effect of the knockout was observed in the development of pancreatic progenitors *in vitro*, not affecting lineage specification upon orthotopic transplantation *in vivo*. The modest effect is in line with the finding that genetic variants near *PPDPF* are associated with random glucose levels in humans, but not with type 2 diabetes risk, supporting that dysregulation of this gene may only result in minor alterations of glycaemic balance in humans. In addition, PPDPF is less organ- and cell type specifically expressed in higher vertebrates and its so far reported functions appear highly context-dependent.

## Author summary

Human pluripotent stem cells (hPSCs) care capable of differentiating into nearly all cells of the human body and thereby serve in research as a powerful tool to study developmental processes in humans. In this study, we explored the role of *PPDPF*, a poorly

**Data availability statement:** The values used to build graphs are compiled in S6 Table. Own RNA-seq data (FASTQ file) and pre-processed files (read count and R log transformed count matrix) are deposited at Gene Expression Omnibus (GSE282625). Publicly available data include ATAC-seq of CyT49 (GSE149148 [18]), ATAC-seq of HUES8 (GSE167606 [13]), histone ChIP-seq (H3K4me1, H3K27ac, and H3K4me3 (GSE149148) and transcription factor ChIP-seq. PP ChIP-seq of FOXA1, FOXA2, SOX9, GATA4, GATA6, PDX1 were retrieved from (GSE149148), NKX6-1 from (GSE167606), the differentiation time course of FOXA1, FOXA2 from (GSE54471 [17]) and murine E17.5 ChIP-seq from (GSE47459 [75]). Codes and data to run codes are also available under https://github.com/MarkusBreunig1/PPDPF-is-not-a-key-regulator-of-human-pancreas-development. Additional resources such as plasmids and generated cell lines are available from the corresponding authors on reasonable request, provided that national and international regulations (e.g. about the work with human embryonic stem cells) are met.

**Funding:** Main funding is provided to A.K. and S.L. by the Else Kröner Fresenius Foundation (EKFS) "2011_A200". A.K. was additionally supported by a Boehringer Ingelheim Ulm cooperation fund and by the DFG (K.L. 2544/7-1) and the Baden-Württemberg Stiftung GmbH („ISF-015 CrossIngPanC"). S.L. received supportive funds by the DFG (LI 2044/5-1 and LI 2044/5-2), M.B. by the intramural Bausteinprogram of Ulm University (L.SBN.0207) and by the EKFS "2022_EKEA.89" and C.J.G. by the Helmholtz-Gemeinschaft (iMed—the Helmholtz Initiative on Personalized Medicine). The funders had no role in study design, data collection and analysis, decision to publish, or preparation of the manuscript.

**Competing interests:** The authors have declared that no competing interests exist.

understood gene linked to pancreas development. While the zebrafish equivalent of this gene, *exdpf*, plays a critical role in forming the exocrine pancreas, its developmental function in mammals, including humans, had not been studied in detail. By using CRISPR/Cas9 technology, we "knocked out" the *PPDPF* gene in hPSCs and directed these cells to form pancreas-specific cells. Interestingly, unlike zebrafish, the "knock-out" of *PPDPF* was not essential for proper pancreatic cell type specification in human cells. Our findings underscore the importance of studying genes across different species to understand their evolutionary and functional nuances. The role of *PPDPF* appears to be more specific and critical in zebrafish, while in humans, it may rather act as a subtle regulator rather than a key driver. This less important role aligned with genetic data suggesting that variations in the *PPDPF* gene influence glucose levels but are not significantly associated with the risk to develop diabetes.

## Introduction

95% of the pancreas is composed of exocrine tissue with acinar glands producing and secreting digestive enzymes into a ductal network from which the pancreatic fluid is transported in the duodenum for processing of food. The remaining 5% of the pancreas is represented by endocrine islets of Langerhans, a specified tissue that is interspersed throughout the pancreas and is crucial for the regulation of blood glucose homeostasis. While physiological functions of the exocrine and endocrine pancreas are highly distinct, both compartments develop from a common ancestor, the multipotent pancreatic progenitor (PP) cells. Understanding embryonic and fetal pancreatic development, as well as the underlying molecular gatekeepers of exocrine and endocrine specification, is highly important for (i) studying causes of pancreatic agenesis, (ii) identifying genetic risk factors of diabetes, (iii) improving pancreatic *in vitro* differentiation strategies for regenerative medicine, and (iv) modelling endocrine and exopcrine pancreatic diseases.

*Exdpf* has been identified in an RNA in situ hybridization screen as an essential regulator of exocrine pancreas development in zebrafish [1]. While the knockdown of *exdpf* nearly completely abolished exocrine cell development, its overexpression increased exocrine and reduced endocrine cell mass suggesting a critical and gatekeeping role during exocrine and endocrine lineage segregation [1]. Mechanistically, *exdpf* was found to act downstream of *ptf1a* and retinoic acid signalling [1]. Upon ablation, effectors of growth restriction such as *p21* were increased, and proliferation of exocrine cells was decreased [1]. This closely related function of *exdpf* for lineage specification and proliferation has drawn considerable attention to the human ortho-logue PPDPF for tumorigenesis. Indeed, several studies have revealed a pro-tumorigenic role of PPDPF in non-small cell lung cancer (NSCLC) [2–4], in ovarian cancer [5], in colorectal cancer [6] and in pancreatic ductal adenocarcinoma (PDAC) [7]. For hepatocellular carcinoma (HCC), findings were controversial. While a more recent study including explorative and mechanistical data found that decreased PPDPF expression correlated with poorer prognosis of HCC tumors [8], an initial HCC study postulated poorer survival in case of increased PPDPF expression [9]. In another study, PPDPF was utilized as part of a three-gene marker panel for biochemical recurrence of prostate cancer after treatment [10].

Studies in the zebrafish have greatly enhanced our understanding of endoderm development and organogenesis but species-specific differences have been also delineated lately [11]. The substantial degree of species conservation [1] together with the emerging literature on the role of PPDPF for tumor growth supported the hypothesis of a potentially important function of human PPDPF during pancreas development. Within this project, we now investigated

whether the human orthologue PPDPF indeed plays an important role for human pancreas development. Implementing a systematic and comprehensive analytical pipeline, we however demonstrate that PPDPF is not essential for proper specification of exocrine and endocrine tissue in a human pluripotent stem cell (PSC)-based differentiation model underpinning essential differences between the pancreas development of zebrafish and higher vertebrates.

## Results

### PPDPF is expressed in the pancreas and in other organs during mouse and human development

To investigate whether *Ppdpf* is expressed in the pancreas during higher vertebrate development, we performed in situ hybridization (ISH) on mouse fetuses at E14.5. In addition to the pancreas, we detected similar high expression levels of *Ppdpf* in other organs such as the stomach, hindgut, duodenum, and kidney (Fig 1A-C). A relatively broad expression pattern including additional organs such as dorsal root ganglion, umbilical artery, and urogenital sinus appears likely. ISH analysis on abdominal sections, however, did not allow mapping of *Ppdpf* expression throughout all organs. We also enquired the human proteome map database [12], which revealed broad expression of PPDPF across multiple fetal and adult organs including the pancreas (Fig 1D). This clearly contrasts with *exdpf* zebrafish expression showing a more restricted organ-distribution to somites, liver, with the strongest signal in the pancreas [1]. Taken together, PPDPF is expressed in the developing and adult pancreas of mice and humans but with a broader expression pattern across multiple other organs than in zebrafish [1].

### PPDPF is expressed along pancreatic differentiation of hPSCs and is regulated by FOXA1/2 TFs

Our platform to differentiate hPSCs into pancreatic progenitor (PP) cells along definitive endoderm (DE), gut tube endoderm (GTE), and pancreatic endoderm (PE) allows the mimicry of and access to key stages of human pancreas development [13–16] (Fig 2A). While *PPDPF* mRNA expression gradually increased from hPSC to PP stage (Fig 2B), protein expression peaked at PE stage (Fig 2C). To understand how PPDPF expression is regulated, we examined publicly available ATAC-seq and ChIP-seq data of us [13] and others [17,18]. We found a gene regulatory promoter region (H3K4me3$^+$) at the *PPDPF* locus with open chromatin as early as DE (Fig 2D) or PE in our data (S1A Fig). While the promoter was poised (H3K27ac$^-$) at DE, it was active (H3K27ac$^+$) at later pancreatic stages (Figs S1B and 2E). The promoter was bound by FOXA1/2 and SOX9 but not by transcription factors (TFs) being more specific to the pancreas such as PDX1 and NKX6-1 (Fig 2F). The endoderm-specifying TFs FOXA1/2 can regulate gene expression by either rendering the gene locus accessible at early stages for subsequent binding of tissue-specific TFs (pioneering role) or by being recruited themselves from tissue-specific TFs at later stages [19]. Here, FOXA1 and FOXA2 already bound to the PPDPF promoter at DE and GTE stages, respectively (Fig 2G), in line with a pioneering role and its broader gene expression profile across several tissues (Fig 1D). ChIP-seq data at a later time point of murine development confirmed Foxa2 binding, while acinar-specific TFs, Ptf1a and Rbpjl, (additionally) bound a more downstream region (S1C Fig) that might act as putative enhancer.

### Genetic variants near PPDPF are associated with glycaemic control

Since several genes associated with diabetes are involved in PP development, we explored publicly available genetic data for *PPDPF* association with metabolic traits. We explored the *PPDPF* region in HugeAMP [20] and found a credible set of 4 single nucleotide

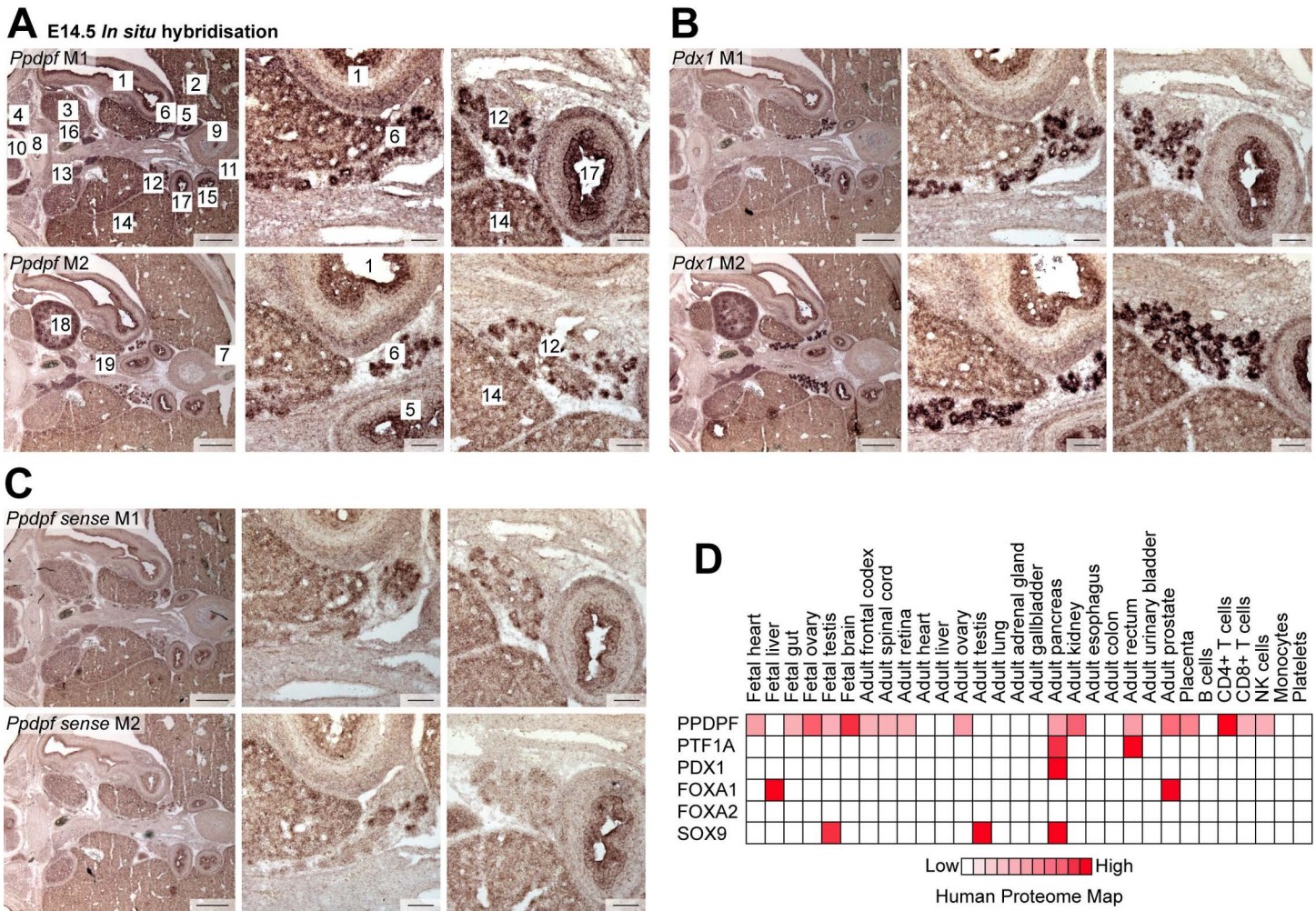

**Fig 1. PPDPF is expressed in the pancreas and in other organs during mouse and human development.** (A-C) *In situ* hybridization on E14.5 mouse fetuses in transverse plane against (A) *Ppdpf*, (B) a positive *Pdx1*, and (C) a negative *Ppdpf sense* control probe. M1: mouse 1; M2: mouse 2. [1] stomach; 2) left lobe of liver; 3) left adrenal primordium; 4) dorsal root ganglion; 5) hindgut; 6) primordium of the left lobe of the pancreas; 7) left umbilical artery; 8) notochord; 9) urogenital sinus; 10) spinal cord; 11) right umbilical cord; 12) primordium of right lobe of pancreas; 13) right adrenal primordium; 14) right lobe of liver; 15) descending part of duodenum; 16) aorta; 17) ascending part of duodenum; 18) left kidney; 19) portal vein). **(D)** PPDPF expression according to the human proteome map database [12] plotted against PTF1A, PDX1, FOXA1, FOXA2, and SOX9 expression profiles.

polymorphisms (SNPs) being associated with increased random glucose (Fig 2H and 2I and S1 Table). The same set was recently reported in a genome-wide association study (GWAS) of random glucose [21]. In addition, traits indicative of renal dysfunction such as increased serum creatine and decreased estimated glomerular filtration rates (eGFR), and other glycaemic traits such as increased fasting glucose and HbA1C levels were significantly associated with the identified SNPs (Fig 2H and S1 Table). Of note, the presence of the affect alleles is associated with decreased PPDPF expression, as measured by cis-expression quantitative trait locus (cis-eQTLs [22]) analysis in blood (Fig 2J and S1 Table). One of these 4 SNPs, rs72629024, is located in the first intron of PPDPF in a binding region of FOXA1/2 and SOX9 (Figs 2D-G, S1A and S1B). Of note, the homologous intron 1 region in zebrafish has been recognized as an important gene regulatory region for ptf1a binding [1] and FORGEdb analysis predicted that this specific SNP might be more important than the other three SNPs in the

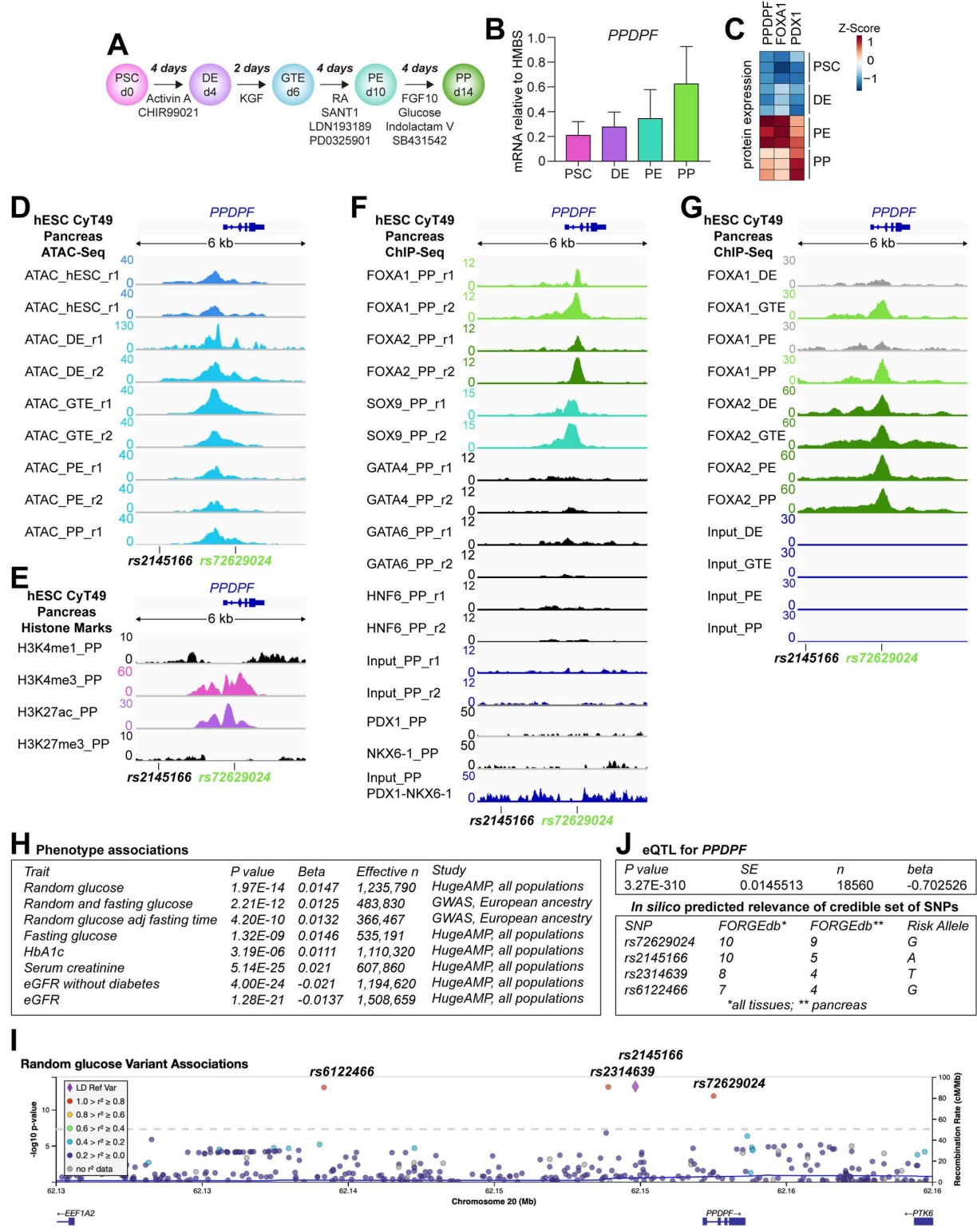

**Fig 2. Regulation of PPDPF expression in human in vitro pancreatic differentiation and correlation to human diseases. (A)** Schematic overview of pancreatic *in vitro* differentiation from human pluripotent stem cells (hPSCs) via definitive endoderm (DE) over gut tube endoderm (GTE) and pancreatic endoderm (PE) into pancreatic progenitors (PP). **(B)** *PPDPF* mRNA expression relative to *HMBS* during *in vitro* differentiation of the human embryonic stem cell (hESC) line HUES8 (n = 4). **(C)** Protein expression during differentiation based on full proteome measurements [23]. **(D)** Assay for Transposase-Accessible Chromatin using sequencing (ATAC-seq)-based chromatin opening at *PPDPF* locus from

publicly available *in vitro* differentiation data of the hESC line CyT49 [18]. **(E)** Histon marks indicating an active promoter region at PP stage in the same CyT49 differentiation. **(F)** Chromatin immunoprecipitation (ChIP)-seq binding peaks of key transcription factors (TFs) at PP stage of CyT49 differentiation [13,18]. **(G)** Time course of ChIP-seq FOXA1/2 binding peaks in CyT49 differentiation [17]. ATAC- and ChIP-seq peaks have been visualized via Integrative Genomics Viewer (IGV) [54]. **(H)** Significant associations of the lead SNP rs2145166 with phenotypic traits. Association data was extracted from the common metabolic diseases knowledge portal of HugeAMP [20]. **(I)** cis-eQTLS [22] correlation analysis between lead SNP and *PPDPF* expression (top). eQTLs: expression quantitative trait locus. *In silico* prediction of the relevance of individual SNPs of the credible set (bottom) using FORGEdb [55]. Localization of the two closer SNPs is displayed in E-H. Note that no chromatin or ChIP-seq peak was found close to the two more distant, not displayed, SNPs. **(J)** Exemplary plot of random glucose trait for lead SNP rs2145166 with linkage disequilibrium (LD) analysis emphasizing that individual effects of the 4 SNPs in the credible set cannot be distinguished (color code) and that these SNPs are the only SNPs close to the *PPDPF* locus that are associated with glycaemic traits.

pancreas (Fig 2J), while linkage disequilibrium analysis could not finally delineate the individual contribution of the 4 SNPs to the actual phenotypic traits (Fig 2I and S1 Table).

## PPDPF-deficient hESCs differentiate into PPs with reduced efficiency

To study the impact of PPDPF during pancreatic differentiation, we employed a four times CRISPR/Cas9 nicking strategy in the human embryonic stem cell (hESC) line HUES8 (S2A Fig). Based on PCR and sequencing (S2B and S2C Fig), we selected two clones harbouring a homozygous deletion of 63 bps of exon3 in the coding sequence. qPCR and Western Blotting revealed loss of transcript and protein expression of the PPDPF$^{p.R19\_I40delInsL/p.R19\_I40delInsL}$ clones (S2D-F Fig). Expression of pluripotency marker remained unaltered upon gene editing (S2G-H Fig). The two clones are from now on referred to as PPDPF$^{KO/KO}$ cl. 1 and cl. 2. While PPDPF-deficient hESCs differentiated into KIT$^+$/CXCR4$^+$ DE and PDX1$^+$ PE with similar efficiencies as their wildtype (WT) controls, the capacity to generate PDX1$^+$/NKX6-1$^+$ PPs was significantly diminished upon ablation of PPDPF (Fig 3A and 3B). Flow cytometry results could be confirmed by immunofluorescence staining (Fig 3C) and a panel of additional qPCR marker including *GATA4* and *GATA6* at DE stage, *SOX9*, *PROX1*, and *FOXA1* at PE stage and *PTF1A*, *GP2*, and *CPA1* at PP stage (Fig 3D).

## Identification of interaction partners not including pancreatic TFs

To further dissect this observation, we screened for potential interaction partners of PPDPF using a PPDPF-Flag overexpression SILAC-screen (stable isotope labeling by amino acids in cell culture) in HEK293T cells followed by mass spectrometry. Indeed, we detected 191 PPDPF-bound proteins from which 45 reached significance across four experiments (Fig 4A and S2 Table). Next, we clustered the list of putative interaction partners along a proteome time course of pancreatic *in vitro* differentiation [23] to obtain a context-relevant protein set followed by overrepresentation analysis of PE- and PP-specific proteins. While PE-specific interaction partners correlated with RNA binding (18 from 22 in query), nucleus [21], and mRNA splicing terms [14], PP-specific interaction partners were linked to protein binding (48 from 49 in query), cytoplasm (46 from 49), and cytoskeleton (17 from 49) terms (Fig 4B and 4C). We proved the validity of the identified interaction partners by exemplary validation of one significant hit (MAGED1) and one non-significant hit (KCTD12) by co-immunoprecipitation experiments and Western Blotting (Fig 4D and 4E).

Due to the prominent role of the zebrafish orthologue *exdpf* for pancreatic development, we had hypothesized a central function of PPDPF as signaling protein in the TF circuitry of the pancreas. Since no pancreas-specific TF could be identified in our HEK293T cell screen, we additionally analyzed the interaction of PPDPF to the pioneering TFs GATA4, FOXA1, and FOXA2 and the key pancreas determining TFs PDX1 and NKX6-1 but did not detect binding to any of the investigated TFs (Fig 4D and 4E).

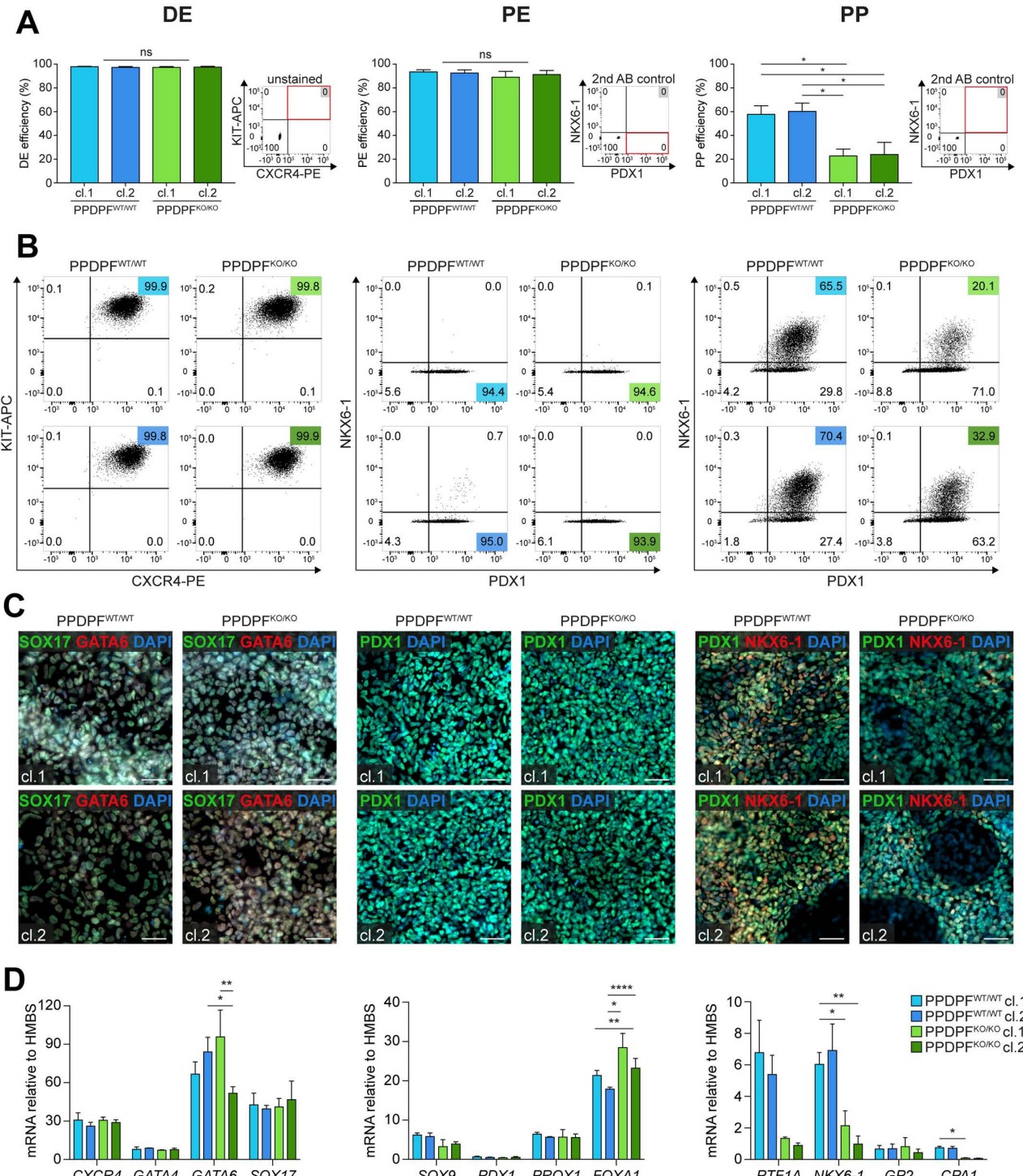

**Fig 3. PPDPF-deficient hPSCs differentiate into PPs with reduced efficiency. (A)** Differentiation efficiencies of clonal PPDPF^WT/WT and PPDP-F^KO/KO hPSC lines into DE, PE, or PP cells, measured by flow cytometry (DE = 6, PE = 4, PP: n = 5 (independent differentiations), in duplicates (two wells per differentiation); Mean ± SEM; ordinary one-way ANOVA, not assuming sphericity, followed by Tukey's multiple comparison test. **(B)** Corresponding flow cytometry plots. **(C)** Immunocytochemistry/immunofluorescence staining confirming flow cytometry results at DE (left), PE (middle), and PP stage (right panel) (Scale bar: 50 μm). **(D)** Stage-specific (DE, PE, PP) mRNA expression relative to *HMBS* (n = 3; Mean ± SEM; ordinary two-way ANOVA, not assuming sphericity, followed by Tukey's multiple comparison test).

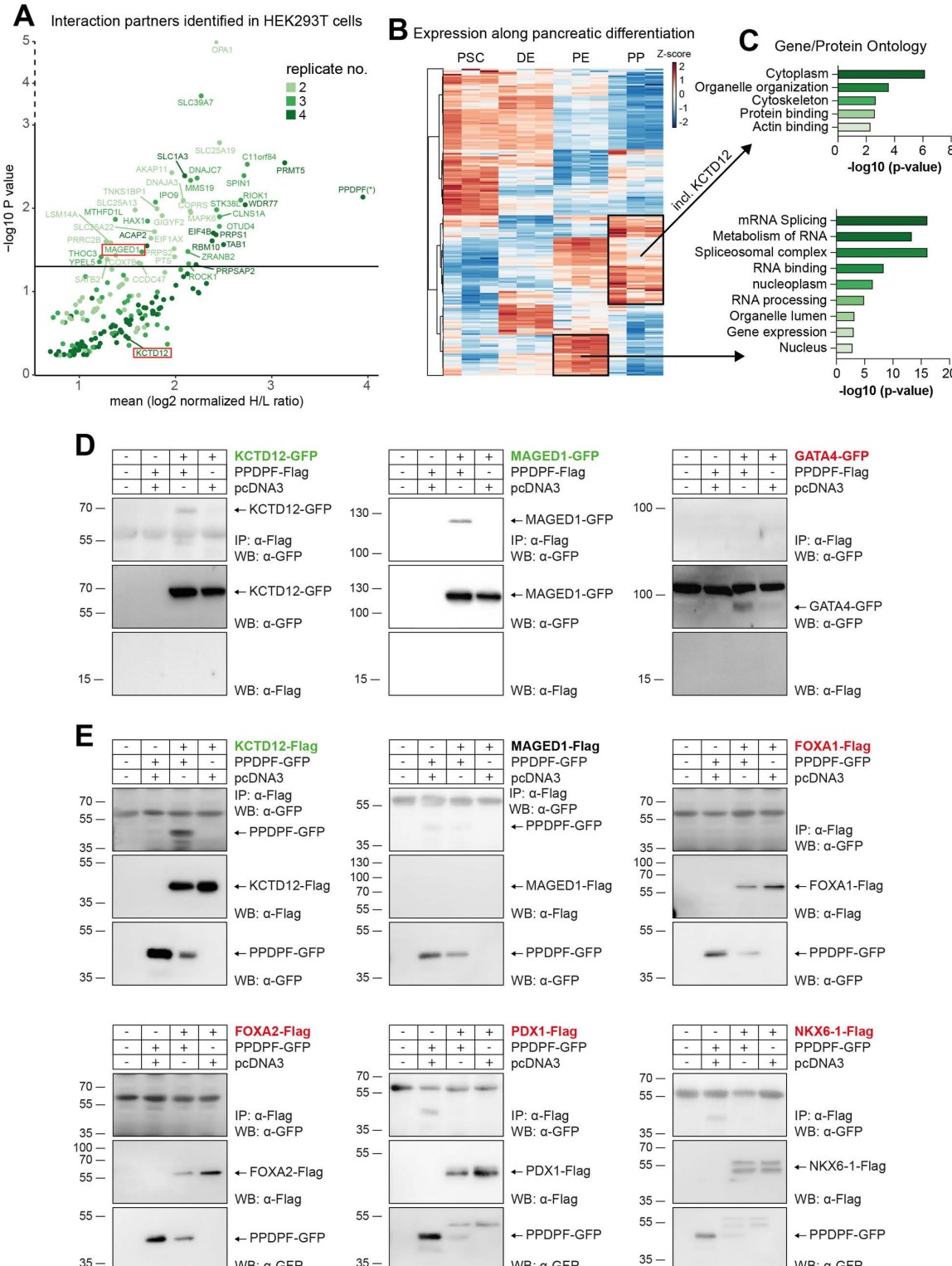

**Fig 4. Identification of interaction partners not including pancreatic TFs. (A)** Potential PPDPF interaction partners identified by immunoprecipitation experiments. PPDPF overexpression in HEK293T cells was followed by stable isotope labeling by amino acids in cell culture (SILAC) and mass spectrometry. H/L ratio: heavy/light isotope ratio. P-value of 0.05 is indicated by a horizontal line; n = 4, with color code illustrating in how many screens individual proteins were detected. **(B)** Expression of potential interaction partners is plotted along pancreatic differentiation using our previous proteome data [23]. Ward.D2 clustering identified PE and PE/PP-specific

proteins (**C**) that were applied to gProfiler overrepresentation analysis. (**D**) Anti-Flag co-immunoprecipitation blots after overexpression of PPDPF-Flag bait and either KCTD12-, MAGED1, or GATA4-GFP in HEK293T cells. (**E**) As in (D) but PPDPF-GFP was overexpressed together with either KCTD12-, MAGED1-, FOXA1-, FOXA2-, PDX1-, or NKX6-1-Flag.

## RNA-seq analysis suggests incomplete PP differentiation and tissue bifurcation upon PPDPF ablation

To globally map developmental consequences of PPDPF ablation, RNA-seq experiments were performed from 2 PPDPF$^{WT/WT}$ and 2 PPDPF$^{KO/KO}$ clones at hPSC, DE, PE, and PP stage from four independent differentiations. First, we utilized global transcriptome data to check for potential off-target effects causing strong transcript expression changes after gene editing but found none in CRISPOR [24] predicted risk candidates (S3A-E Fig). RNA-seq samples clustered according to the developmental time point in principal component analysis (PCA) of the 10% most variant genes (Fig 5A), with a subtle separation of PPDPF$^{WT/WT}$ and PPDPF$^{KO/KO}$ samples at PE and PP stage (Fig 5B). In hierarchical clustering, 4 of 8 KO samples clustered closer to PE samples than to the other PP samples suggesting an incomplete differentiation into PPs upon PPDPF ablation (Fig 5C).

While very few genes were deregulated at hPSC and DE stage, around 300 differentially expressed genes (DEGs) were found at PP stage (Fig 5D and S3 Table). GO-term analysis of DEGs upregulated in PPDPF$^{KO/KO}$ samples pointed to a bifurcation towards non-pancreatic tissues with gene terms for gut enterocytes and lymphoid cells being overrepresented (Fig 5E and S4 Table). In addition, terms for pancreatic precursor cells and ECM-receptor interaction were downregulated (Fig 5E and S4 Table). Venn diagrams, depicting the top 10 DEGs, illustrated that several of these were altered at more than one developmental stage (Fig 5F). Among them the lncRNA H19 that was found to have diverse roles in proliferation and tumorigenesis by sponging microRNAs [25] was for example downregulated, while ZIM2 and PEG3 that can also regulate proliferation and tumorigenesis but also fetal growth [26] were upregulated in PPDPF$^{KO/KO}$ cells. We did however not detect specific pathways to be globally affected by loss of PPDPF. Although some pathways such as E2F targets and G2M checkpoints were upregulated in PPDPF$^{KO/KO}$ samples at PE stage (Fig 5G), (i) alterations did not persevere to PP stage (Fig 5G), and (ii) were likely rather caused by fluctuations of single samples instead of a consistent genotype-specific pattern (Fig 5H).

## Transcriptomic changes between genotypes are relatively small

To approach a potential developmental defect more specifically, we compiled tailored gene sets for hPSC-based pancreatic *in vitro* differentiations [23,27], human fetal tissue [28–30], and Ptf1a targets [31] (S5 Table). Gene set enrichment analysis substantiated our previous findings that PPDPF$^{KO/KO}$ PE/PP cells were depleted for a pancreatic progenitor identity while they were enriched for an endodermal identity (Fig 6A). Conclusively, genes that were differentially upregulated upon Ptf1a$^{KO/KO}$ in mice, were depleted in PPDPF$^{KO/KO}$ cells (S4A Fig). In addition, human fetal pancreatic signatures including the more mature tip and trunk states were depleted (S4B and S4C Fig), while several non-pancreatic tissue signatures such as intestine and placenta were enriched (S4D Fig). Related to latter, several pregnancy-specific glycoproteins (PSGs) were significantly upregulated in PPDPF$^{KO/KO}$ PE/PP samples, but their actual transcript levels remained extremely low (S4D Fig and S3 Table). Heatmap illustration of enriched and depleted gene sets, however, revealed that also PPDPF$^{KO/KO}$ cells have successfully acquired stage-specific signatures including a pancreatic progenitor expression profile and that differences between stages are much higher than between genotypes (Fig 6C and 6D).

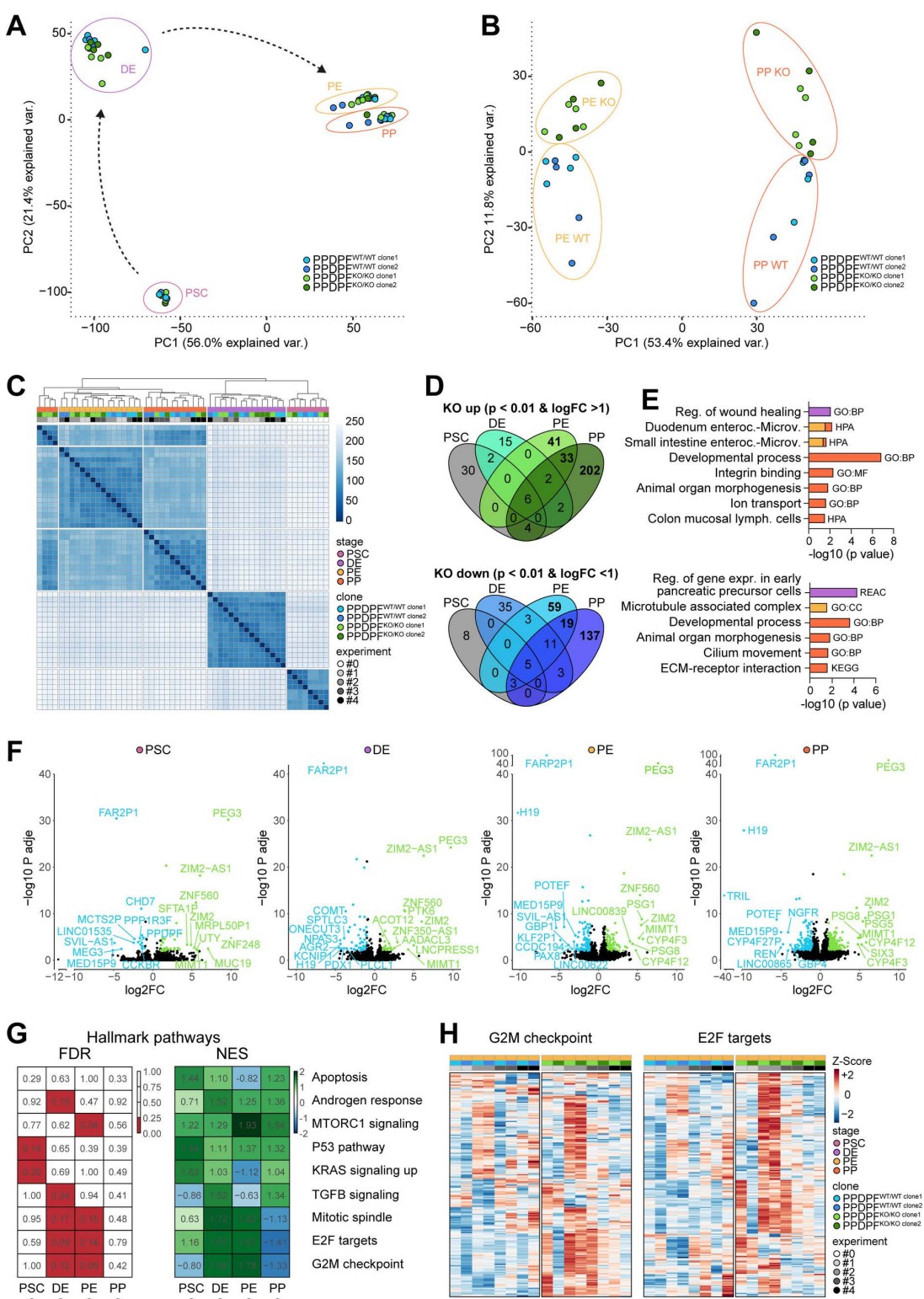

**Fig 5. RNA-seq analysis along pancreatic progenitor differentiation.** (A) Principal component analysis (PCA) of top 10% most variant genes along the time course of pancreatic differentiation (hPSC: n = 2; DE-PP: n = 4). (B) PCA of the PE and PP sample subset. (C) Hierarchical clustering of all genes using Euclidean distance calculation. (D) Venn diagram of stage-resolved differentially expressed genes (DEGs). (E) Overrepresentation analysis of DEGs using gProfiler [56]. Color scheme highlights

differentiation stage (based on legend in C). Gene sets in top panel were enriched in PPDPF$^{KO/KO}$ cells, bottom panel were depleted. **(F)** Volcano plots of stage-specific PPDPF$^{KO/KO}$ versus PPDPF$^{WT/WT}$ gene expression comparison. DEGs that are up in PPDPF$^{KO/KO}$ samples are highlighted in green, genes that are down in blue. Top ten DEGs with highest FC change are labelled. **(G)** Gene set enrichment analysis (GSEA) against hallmark pathways. False discovery rates (FDRs) ≤ 0.25 are highlighted in red (left panel), while comparisons that are enriched in PPDPF$^{KO/KO}$ cells are depicted in green and comparisons that are depleted in blue (right panel). Additional information: For "G2M checkpoint", 0 out of 197 genes were significantly upregulated (DEGs) in PPDPF$^{KO/KO}$ PPs over PPDPF$^{WT/WT}$ PPs, while 1 was downregulated. For "E2F targets" 0 out of 200 genes were up or down. **(H)** Heatmap illustrating row/gene-scaled gene expression at PE stage of two selected hallmark pathways. PSC: pluripotent stem cell; DE: definitive endoderm; PE: pancreatic endoderm; PP: pancreatic progenitor.

Indeed, only 19 of 421 genes within the "Definitive endoderm [23]" gene set were significantly upregulated (DEGs) in PPDPF$^{KO/KO}$ PPs over PPDPF$^{WT/WT}$ PPs, while 5 were downregulated (P-value < 0.01 and FC > I1I). At the same time 5 of 451 genes within the "Pancreatic progenitor [23]" gene set were upregulated (DEGs) and only 15 downregulated in PPDPF$^{KO/KO}$ PPs (S3 and S5 Tables). Time-resolved gene expression profiles of key endoderm (Fig 6E) and pancreatic progenitor genes (Fig 6F) further highlights that transcriptomic changes were overall rather small.

## No alteration in proliferation and growth characteristics of PPDPF$^{KO/KO}$ PPs

Since *exdpf* plays a role in the proliferation of exocrine cells in zebrafish [1], we specifically checked the effect of PPDPF ablation on proliferation and growth characteristics of pancreatic precursors. The total number of cells during pancreatic *in vitro* differentiation was however not reduced in PPDPF$^{KO/KO}$ cells (Fig 7A). Also, the rate of proliferation and the cell cycle distribution was not significantly reduced (Fig 7B-D). A slight trend of a reduction in S-phase was visible in PPDPF$^{KO/KO}$ PPs but did not manifest in changes in absolute cell numbers (Fig 7A). In addition, the number of cleaved CASP3$^+$ or Annexin V$^+$ cells was not altered (Fig 7E and 7F), and also global RNA-seq analysis did not suggest a change in growth- and cell death-related programs (Figs 7H, 5G and 5H). In summary, our data does not support a major impact of PPDPF on proliferation and growth of human pancreatic progenitors.

## PPDPF is broadly expressed across fetal pancreatic cell types beyond PP stage

Until now, we observed a reduction in PPs in *in vitro* differentiation experiments of PPD-PF$^{KO/KO}$ hESCs and RNA-seq experiments revealed the preservation of endodermal gene expression and a lineage bifurcation into non-pancreatic tissue, however, both at relatively low expression level. To investigate how PPDPF is expressed beyond PP stage, we reanalysed a human fetal pancreas scRNA-seq data set covering key stages from PP formation, trunk and tip development until cell type specification (Fig 8A and 8B) [32] and found that PPDPF is broadly expressed at all investigated time points 4 to 11 weeks post conception (W4-11) (Fig 8C and 8D). While PPDPF expression increased from PP/MP stage to tip, trunk and endocrine precursors (EP), it remained expressed in all fetal pancreatic cell types (Fig 8C and 8D). Of note, albeit expressed, PPDPF expression was lowest in fetal acinar cells contrasting the exocrine-specific expression and function of its *exdpf* orthologue during zebrafish development (Figs 8C, 8D and S5A-H). In addition, we investigated PPDPF expression in an adult pancreatic scRNA-seq data set [33], where PPDPF appears also broadly expressed in different adult pancreatic cell types (S5I Fig).

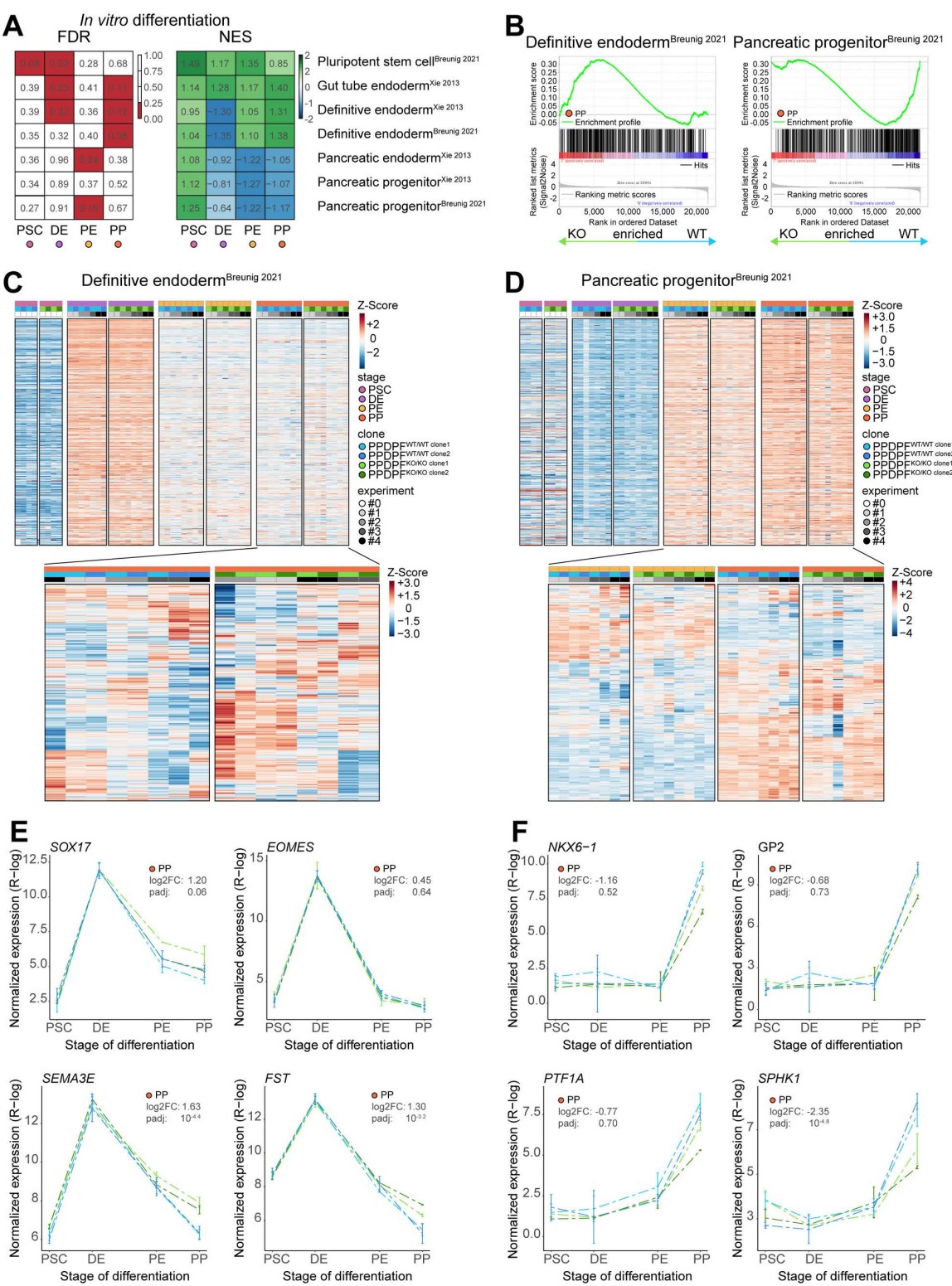

**Fig 6. Incomplete pancreatic progenitor differentiation upon PPDPF ablation.** **(A)** Gene set enrichment analysis (GSEA) against gene sets compiled from pancreatic differentiation experiments [23,27]. False discovery rates (FDRs) ≤ 0.25 are highlighted in red (left panel), while comparisons that are enriched in PPDPF[KO/KO] cells are depicted in green and comparisons that are depleted in blue (right panel). **(B)** GSEA plots against a definitive endodermal and pancreatic progenitor gene set are shown for the comparison of PPDPF[WT/WT] and PPDPF[KO/KO] PPs. **(C)** Heatmap illustrating row/gene-scaled gene expression at all stages and PP stage only (Zoom,

bottom panel) of all detected genes within the "Definitive Endoderm" gene set. **(D)** As in C but zoom (bottom panel) shows PE and PP stage, and "Pancreatic Progenitor" gene set is depicted. **(E,F)** Line plots (Mean ± SEM) against key DE (SOX17, EOMES) and PP genes (NKX6-1, GP2, PTF1A) together with exemplary genes top ranked in GSEA (DE: SEMA3E, FTS; PP: SPHK1).

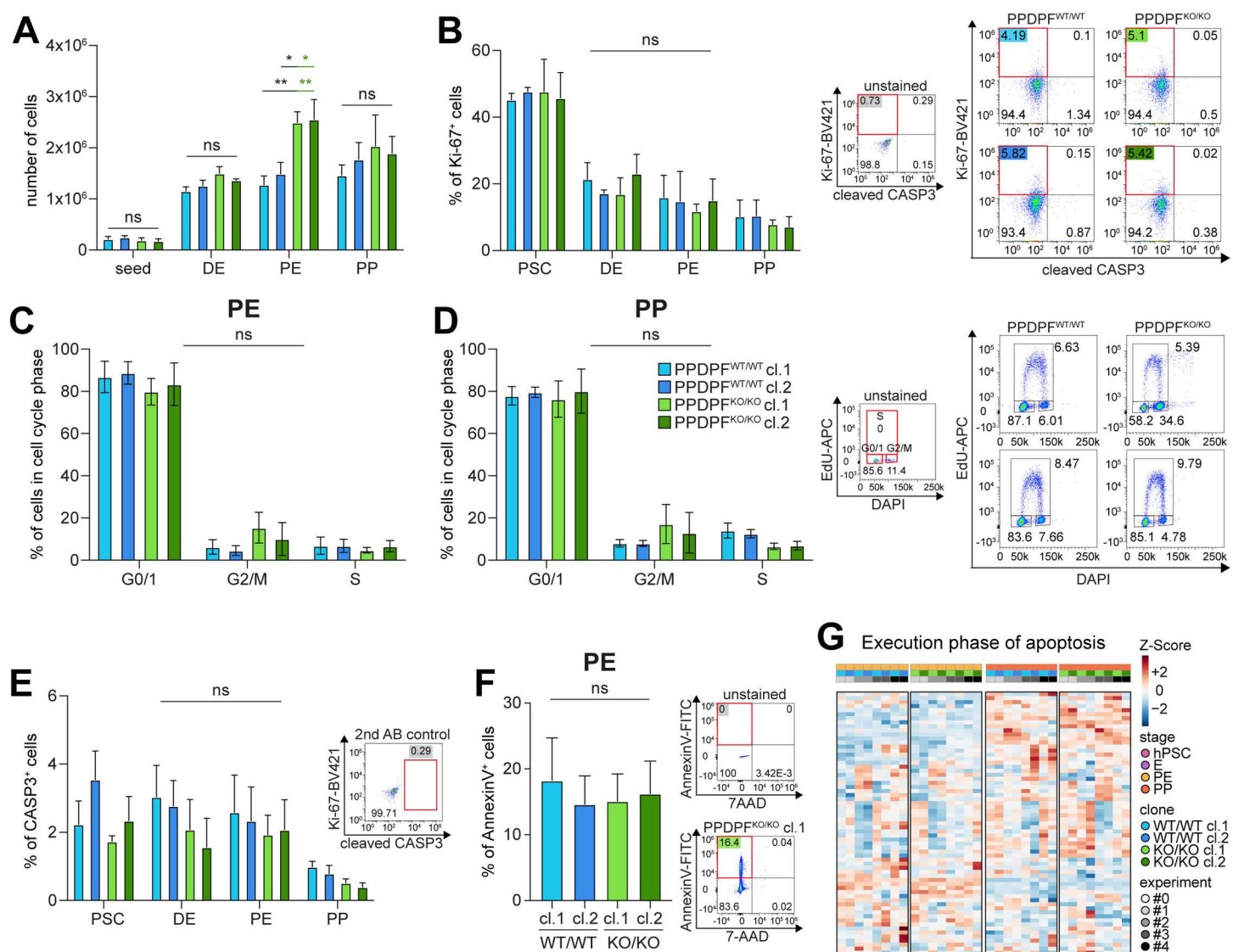

**Fig 7. No alteration in proliferation and growth characteristics of PPDPF[KO/KO] PPs. (A)** Cells per well of a 24-well plate counted with help of a Neubauer chamber. **(B)** Flow cytometry of proliferation marker Ki-67 and gating strategy. **(C-D)** EdU-based cell cycle analysis at PE (C) and PP (D) stage and gating strategy. **(E)** Flow cytometry of apoptosis marker cleaved Caspase 3 (CASP3). For A-E: n = 3, in duplicates; Mean ± SEM; ordinary two-way ANOVA, not assuming sphericity, followed by Tukey's multiple comparison test. **(F)** Flow cytometry of Annexin V and gating strategy. n = 4, in duplicates; Mean ± SEM; ordinary one-way ANOVA, not assuming sphericity, followed by Tukey's multiple comparison test. **(G)** Heatmap of RNA-seq data illustrating row/gene-scaled gene expression at all stages of all detected genes within the hallmark gene set "Execution phase of apoptosis". Additional information: 0 out of 156 genes were significantly upregulated (DEGs) in PPDPF[KO/KO] PE cells over PPDPF[WT/WT] PE cells, while 1 was downregulated. At PP stage, 1 was up and 0 were down.

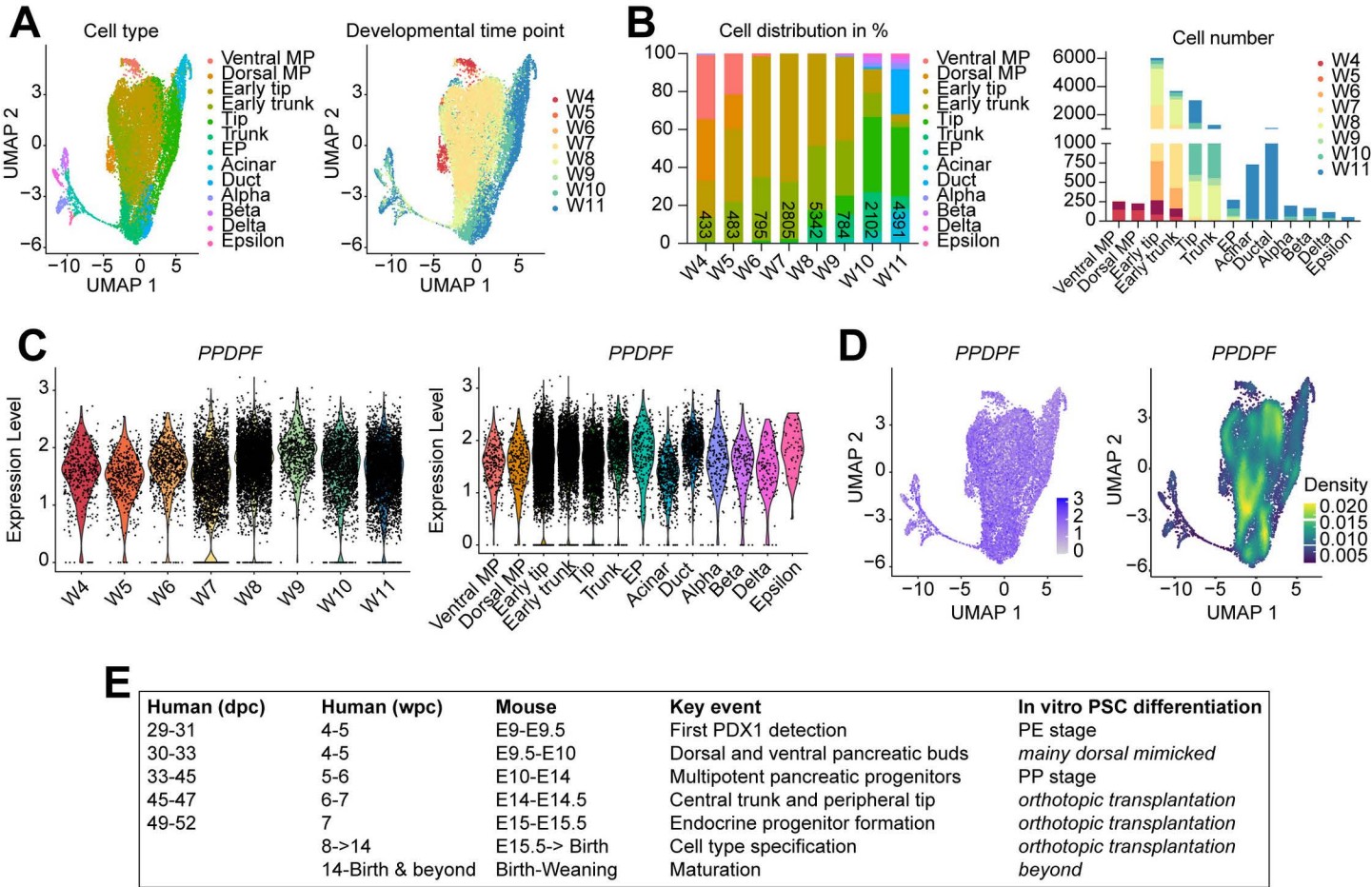

**Fig 8. PPDPF is broadly expressed across fetal pancreatic cell types beyond PP stage. (A)** Reanalysis of a publicly available human fetal pancreas scRNA-seq data set from Ma, Zhang [32]. UMAP representation with cell type annotation (left) along developmental time points (right). **(B)** Number of analyzed cells and time-resolved cell type distribution. **(C)** Violin plots of *PPDPF* expression along developmental time points (left) and within cell types (right). **(D)** *PPDPF* expression within UMAP space. **(E)** Classification of fetal stages of human and mouse pancreas development and estimation of appropriate stage of hPSC-based pancreatic differentiation. Information about early human stages (29-52 dpc) was compiled from Jennings, Berry [41], later stages from Nair and Hebrok [57]. Dpc: days post conception; wpc: weeks post conception; EP: endocrine progenitor; MP: multipotent progenitor; UMAP: Uniform Manifold Approximation and Projection.

## PPDPF^KO/KO cells successfully develop into pancreatic tissue upon orthotopic transplantation

To experimentally investigate the role of PPDPF during pancreatic cell type specification *in vivo* (Fig 8E), we used our previously licensed orthotopic transplantation approach [14,23,34,35] using PE cells derived from 1 PPDPF^WT/WT clone and 2 PPDPF^KO/KO clones into 3 immunodeficient NOD/SCIDgamma (NSG) mice for each condition. Differentiation efficiencies of transplanted PE cells were very similar between genotypes, and we genetically confirmed genotypes of clones at the time point of transplantation as additional quality control (Fig 9A). All 9 engraftments efficiently formed fetal pancreas-like tissue (Fig 9B) of human origin (Fig 9C). While consecutive sections and engraftments were distinct in size and representation of tubular structures, we did not observe genotype-specific alterations. To determine proper lineage segregation, we quantified the expression of key acinar, ductal, endocrine, non-pancreatic and proliferation-related marker in engraftments (+++, ++, +, −)

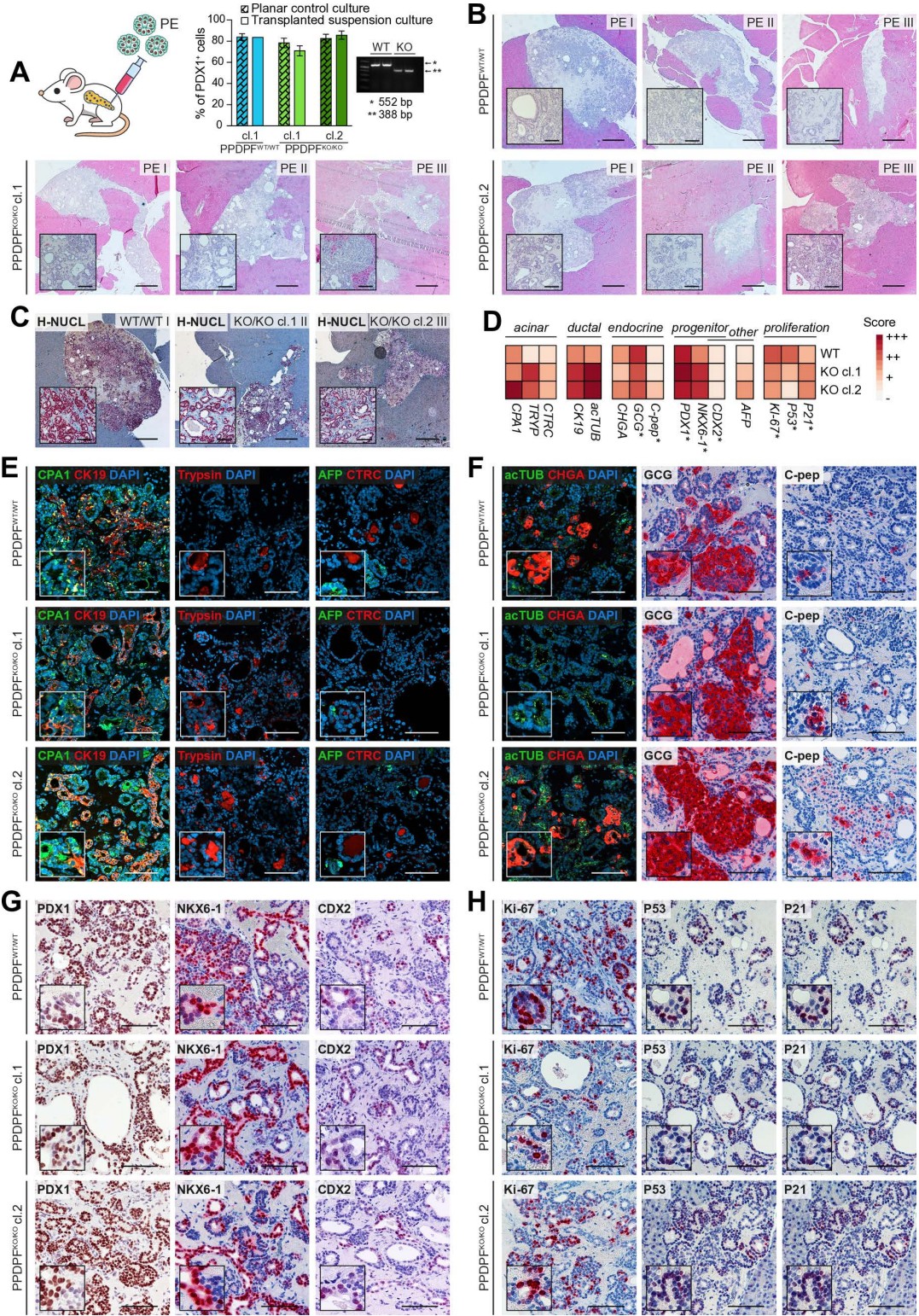

**Fig 9. PPDPF$^{KO/KO}$ PE cells successfully develop into pancreatic tissue upon orthotopic transplantation. (A)** PE cells, that were cultured 2 days in suspension (d7-d9) for improved engrafting, were orthotopically transplanted in the pancreas of NSG mice and propagated *in vivo* for 8 weeks. Quality control of transplanted cells highlighting similar differentiation efficiencies between genotypes and to normal planar PP control (n = 1, in duplicates; Mean ± SD). DNA has been isolated directly prior to transplantation for an additional genotyping PCR control. **(B)** Hematoxylin and eosin (H&E) staining of

identified grafts with n = 3 for PPDPF$^{WT/WT}$ and PPDPF$^{KO/KO}$ cl.1 and cl.2 PEs. **(C)** Human nucleoli (H-NUCL) staining illustrates human engraftment. **(D)** Quantification of depicted marker expression averaged over individual grafts of each genotype (– completely or nearly completely absent; + weak expression; ++moderate expression; +++ strong expression. **(E)** Immunofluorescence (IF) staining of acinar marker CPA1, Trypsin, and CTRC, ductal marker CK19, and liver marker AFP. **(F)** IF and immunohistochemistry (IHC) of ductal cilia marker acetylated tubulin (acTUB), endocrine marker CHGA, GCG, and C-peptide (C-pep). **(G)** IHC of progenitor marker PDX1, NKX6-1, and CDX2. **(H)** IHC of proliferation marker Ki-67 and effectors of growth restriction P53 and P21. For all: Images are depicted that show strong marker expression to highlight trilineage competence. Scale bar: B-C: 500 μm, inlet: 100 μm. E-H: 100μm, inlets 2x enlarged.

(Fig 9D). We neither observed a reduction in acinar digestive enzymes (CPA1, TRYP, CTRC) nor in tubular ductal marker (CK19, acTUB) nor in endocrine (CHGA), alpha (GCG), and beta (C-pep) marker in PPDPF$^{KO/KO}$ grafts (Fig 9D-F). In addition, progenitor marker (PDX1, NKX6-1, CDX2) or non-pancreatic marker (AFP) were also not increased in PPDPF$^{KO/KO}$ grafts (Fig 9G). Finally, we also did not detect a genotype-consistent upregulation of cell-cycle checkpoint effectors (P53, P21) and a reduction of proliferation (Ki-67) (Fig 9H). In summary, PPDPF$^{KO/KO}$ PE cells differentiated efficiently into fetal pancreas-like tissue with similar size of engraftments and similar representation of acinar, ductal, and endocrine-specified cells as PPDPF$^{WT/WT}$ cells. In addition, we performed transplantation experiments of GP2 sorted/ unsorted PPs according to our recently fine-tuned PP protocol [34] (S6A Fig). Importantly, PP transplantations confirmed the competence of PPDPF$^{KO/KO}$ cells to successfully differentiate into pancreatic-like tissue in a highly similar manner as PPDPF$^{WT/WT}$ cells (S6B-H Fig). Of note, the small observed changes in the quantification were not consistent between PE and PP transplantations (Figs 9D and S6D), underlining the preserved capacity of PPDPF$^{KO/KO}$ hPSCs to develop into the most important human pancreatic cell types *in vivo*.

### PPDPF$^{KO/KO}$ cells efficiently differentiate into endocrine islet-like organoids *in vitro*

In a nutshell, observed differences between PPDPF proficient and deficient cells to form PP cells did not perpetuate to more mature pancreatic stages *in vivo*. It remained, however, unclear when PPDPF$^{KO/KO}$ PPs might had caught up to PPDPF$^{WT/WT}$ during PP specification. To dissect temporal aspects more closely, we differentiated PPs *in vitro* into endocrine islet-like organoids [36,37] in a proof-of-concept experiment over 34 days (Fig 10A). hPSCs were first differentiated into PP cells, where the decrease in NKX6-1$^+$ PPs upon PPDPF ablation was again reflected (Fig 10B, left), and no substantial premature activation of endocrine marker was found (Fig 10B, right). In contrast, upon further differentiation into endocrine progenitors (Fig 10C), pre-islets (Fig 10D) and more mature islet-like organoids (Fig 10E), the number of derived alpha-like cells (GCG) and beta-like cells (C-peptide, INS) was highly similar. Comparable endocrine differentiation efficiencies between PPPDF$^{WT/WT}$ and PPDPF$^{KO/KO}$ cells were further validated by immunofluorescence imaging on paraffin-embedded and sectioned islet-like organoids (Fig 10F). We therefore concluded that PPDPF$^{KO/KO}$ PPs harbour the capacity to efficiently specify pancreatic endocrine cells *in vitro* with no differences on marker expression being observed already at endocrine progenitor stage on day 20.

## Discussion

"Nomen est omen", does not appear to keep its promise in case of PPDPF. The gene name *pancreatic progenitor differentiation and proliferation factor* suggests a key role of PPDPF for PP formation and expansion for tissue formation. In our comprehensive hPSC-based *in vitro* and *in vivo* experiments, however, we demonstrated that PPDPF is dispensable for

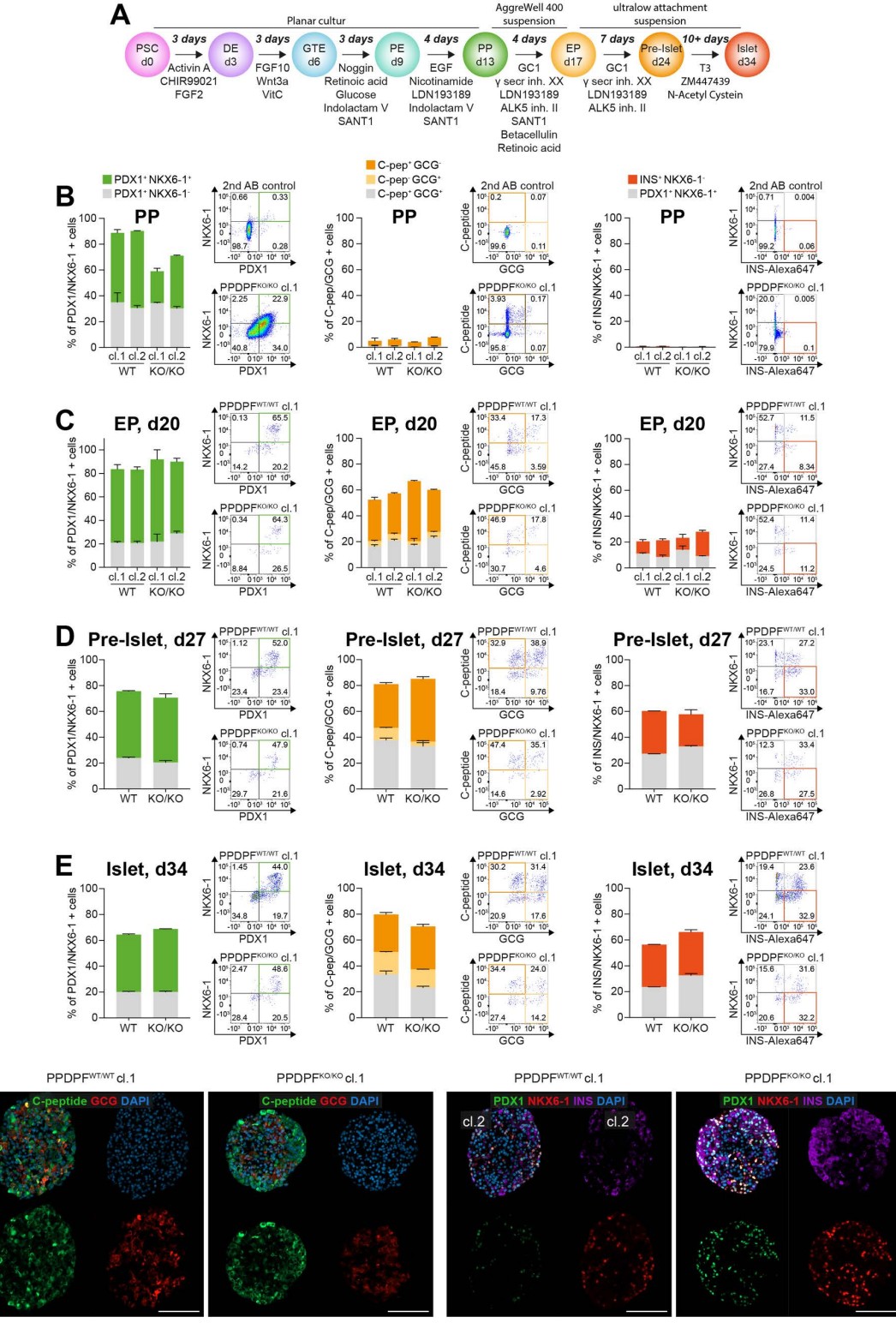

**Fig 10. PPDPF**[KO/KO] **cells efficiently differentiate into endocrine islet-like organoids *in vitro*. (A)** Schematic overview of human pluripotent stem (hPSC) cell-based differentiation into definitive endoderm (DE), gut tube endoderm (GTE), pancreatic endoderm (PE), and pancreatic progenitor (PP) cells in planar cultures followed by 3D endocrine progenitor (EP) formation and maturation into pre-islets and islet-like organoids at d34. **(B-E)** Differentiation efficiencies of clonal PPDPF[WT/WT] and PPDPF[KO/KO] hPSC lines, measured by flow cytometry (n = 1 (independent differentiations), in duplicates (two wells per

differentiation); Mean ± SD; at PP stage (**B**), EP stage (**C**), Pre-islet (**D**) and islet stage (**E**). (**F**) Representative immunofluorescence imaging of key endocrine marker (PDX1, NKX6-1, C-peptide, GCG, INS) on formalin-fixed paraffin-embedded (FFPE) organoid sections at d34 of differentiation. Scale bar: 50 μm.

proper lineage specification and segregation into acinar, ductal, and endocrine cells. While initial *in vitro* differentiation experiments suggested a reduced capacity of PPDPF$^{KO/KO}$ hESCs to develop into PPs, global RNA-seq experiments revealed that stage-specific changes were rather small and might represent a slight delay in PP development together with a slight upregulation of non-pancreatic genes. PE- and PP-specific transplantation experiments and *in vitro* islet differentiations finally disclosed that PPDPF$^{KO/KO}$ hESCs indeed possess the competence to develop into all major cell types of the human fetal pancreas. The second aspect of its gene name suggests an important role of PPDPF for proliferation of PPs. However, proliferation marker and growth characteristics appeared not decreased upon loss of PPDPF during all investigated stages of pancreatic development including *in vitro* differentiation into DE, PE, and PP stage and *in vivo* specification into acinar, ductal, and endocrine cells.

Our data contrasts with the original finding in zebrafish, where the orthologue *exdpf* was found to be a novel player, that is, within the pancreas, specifically expressed in acinar cells, functions as a gatekeeper of exocrine and endocrine segregation, and that is essential for the formation of exocrine pancreatic tissue mass [1]. Utilizing publicly available scRNA-seq and proteome data sets as well as ISH of mouse fetuses, we were able to demonstrate a relatively broad expression profile of murine Ppdpf and human PPDPF throughout development, but also in adult tissue. The lack of cell type and tissue specificity is in line with our observations, where we did not detect an exocrine to endocrine lineage switch upon ablation of PPDPF. Our results might be partly explained by an increasing evolutionary complexity from zebrafish to higher vertebrates and humans. While part of its developmental function might be residually visible in initial *in vitro* differentiation experiments demonstrating an effect on PP generation, the loss of PPDPF might be either compensated by other factors or might be simply not as decisive for the development of the more complex human pancreas.

Specifically, we found that PPDPF expression is regulated by FOXA1/2 and SOX9 TFs at DE to PP stage. FOXA1/2 play a central role for lineage specification and development of the neuronal tube, endodermal organs such as liver, lungs, kidney, prostate, and pancreas [38], the kidney and urethral tube [39]. The lack of binding of PP-specific TFs PDX1 and NKX6-1 fits to the relatively broad expression profile of PPDPF in non-pancreatic tissues. Interestingly, we observed binding of Ptf1a and Rbpjl to an additional distal gene region, likely possessing an enhancer function, at later murine E17.5 stage. While respective ChIP-seq data are not available for human cells or tissues, Ptf1a and Rbpjl are considered highly specific for acinar cells [40]. It remains therefore unclear, how PPDPF expression is regulated at later stages of human development and tissue maintenance. Its broad scRNA-seq expression profile in the pancreas however appears similar to FOXA2 expression (S5 Fig) rendering a sole dependence on PTF1A and acinar-specific function unlikely.

In humans, the pancreas develops from a separate ventral and dorsal anlage and multipotent PTF1A positive PPs give rise to endocrine and exocrine cells [41,42]. Human *in vitro* PP differentiation protocols rather mimic the dorsal pancreas route [29]. Also in zebrafish, ventral and dorsal anlagen can be observed [43] but in contrast to mammals, amphibians and birds, only the ventral but not the dorsal PPs contribute to exocrine pancreas development [44] and ptf1a is only expressed in the ventral pancreas in zebrafish [45]. Since the function of zebrafish *exdpf* was placed downstream of *ptf1a* and since we also noted Ptf1a binding to the Ppdf locus at E17.5, the possibility that our hPSC-based differentiation approach might not

capture a potential phenotype due to the lack of the PTFA1A/ventral pancreas route during PP differentiation, lack of full acinar maturity, or lack of morphogenesis-related processes cannot be completely excluded. However, we consider this rather unlikely, as (i) ventral and dorsal PPs, of which the latter also expresses PTF1A despite of lower expression levels (S5A Fig), contribute to exocrine and endocrine development in humans, (ii) analysis of human fetal development revealed similar PPDPF expression levels in dorsal and ventral PPs not following PTF1A expression (Figs 8C and 5A), and (iii) orthotopic transplantations of PPDPF[KO/KO] hPSC-derived pancreatic cells recapitulated acinar cell specification, secretion of digestive enzymes such as trypsin, and patterning into gland-like structures. In addition, we demonstrate that *PPDPF* expression can be regulated by FOXA1/2 and SOX9, providing an alternative or additional regulation mechanism to PTF1A TF activity.

A recent and large GWAS study specifically focusing on random glucose signals identified 143 SNPs including the PPDPF-associated SNPs highlighted in this study [21]. While the authors exemplarily demonstrated the relevance of such traits for mechanical insights into glucose regulation, occurrence of potential diabetes complications and stratification of diabetes treatment [21], the underlying mechanism how the SNPs that correlate with decreased expression of PPDPF deteriorate glucose homeostasis remained unclear. Within this study, we can now show that one of the credible SNPs (rs72629024) is bound by endodermal FOXA1/2 TFs in a gene regulatory region associated with active promoter marks. The observed reduction in the generation of hPSC-derived PPs could explain changes in glycaemic control, especially in co-occurrence with additional unknown (non-genetic) risk factors. However, effects on glycaemic control can be caused by alterations in multiple organs such as kidney, liver, gut, adipocytes, cardiovascular or neuronal system or even eye and lungs. Hence, also a non-pancreatic function of PPDPF during development or tissue homeostasis could be the underlying cause of the observed glycaemic changes. PPDPF-associated SNPs also correlated with impaired renal activity, and renal and endocrine pancreatic function can affect each other in both directions. A recent GWAS study focusing on kidney function also reported on the correlation of the same PPDPF-associated SNP and impaired renal function [46]. In another study, liver-specific KO of Ppdpf led to fatty liver formation in a mouse model at 32 weeks of age on chow diets, which was enhanced by high fat diet (HFD) [47]. Under both diets, higher blood glucose levels were found in the liver-specific Ppdpf KO mice and Ppdpf overexpression reduced blood glucose levels at least under HFD [47]. The changes in lipid metabolism offer an additional not directly pancreas-related explanation for the correlation between glycaemic alterations and PPDPF-associated SNPs. Importantly, the PPDPF-associated SNPs did not correlate with actual diabetes risk but only with glycaemic alterations thereby supporting the less important role for pancreas development in humans than in zebrafish.

While we initially had hypothesized a role as signaling protein in the developmental TF circuitry, instead, we have identified and exemplarily validated other novel interaction partners of PPDPF by SILAC mass spectrometry after PPDPF overexpression in HEK293T cells. Several of the hits including MAGED1 and KCTD12 have been described in both cancer context and general cellular processes. The list of putative interaction partners generated in HEK293T cells possesses, in our opinion, a valuable resource for future mechanistic studies in the rising field of PPDPF-related cancer studies. Some examples of identified interaction partners, where PPDPF interaction might be potentially involved in context-important gene and protein regulatory functions are USP15, a deubiquitinating enzyme involved in many protein regulatory functions [48], RBM10, a nuclear RNA-binding protein (RBP) that regulates the alternative splicing of primary transcripts [49,50], or the PRMT5-WDR77 complex together with its adapters (CLNS1A, COPRS, RIOK1) involved in methylation of the spliceosome, histones and ribosomal complexes [51] and a promising target of various anticancer therapies

[52,53]. Of note, current literature highly suggests context-specific mechanisms of PPDPF with a putatively converging theme of (indirect) protein modulatory functions [3,4,6–8,47]. We therefore hypothesize that PPDPF acts as a glue-like protein that alters the interaction of bound target proteins to other proteins thereby modulating function and regulation. In agreement, PPDPF is small and appeared very sticky in our SILAC screen experiments.

Furthermore, several studies have disclosed an oncogenic [2–7,9], pro-proliferative [1–3,5] and anti-apoptotic [2–4,8] role of PPDPF, while one study postulated a tumor suppressive and anti-proliferative role [8]. In contrast, we did not detect a pro-proliferative or anti-apoptotic function of PPDPF during our pancreatic differentiation experiments suggesting no cell-conserved and direct function of PPDPF in tissue growth. Instead, the indirect protein regulatory function of PPDPF might only become relevant upon additional inflammatory signals or disturbed cell homeostasis caused by e.g. oncogenic drivers. In conclusion, our study reveals that PPDPF is dispensable for human pancreas development, while current literature suggests an important protein regulatory role of PPDPF during tumor formation and progression in several organs.

## Materials and methods

### Ethics statement

All conducted experiments were carried out in compliance with German law and animal experiments were approved by the governmental office ("Regierungspräsidium") of the state of Baden-Württemberg in Tuebingen, Germany: Xenotransplantation of hESC-derived pancreatic cells into the pancreas of NOD scid gamma (NSG) mice (NOD.Cg-Prkdcscid Il2rgtm-1Wjl/SzJ strain (Charles River); RRID:BCBC_4142) was approved under TVA1406 and mouse organ isolation and processing of E14.5 C57BL/6J embryos under TV-Nr. o.161-3.

### Animal experiments

**Orthotopic transplantation.** NSG mice were used for xenotransplantation of hESC-derived pancreatic cells into the murine pancreas. Mouse were sacrificed 8 weeks after transplantation (56 days for PE transplantation, 52 days for PP transplantation). While the experimental procedure of the transplantation has been detailed previously [23], cell preparation will be described in the pancreatic differentiation section.

*In situ* hybridization (ISH). For riboprobe generation, PCR was performed using GoTaq Flexi Polymerase (Promega) with the following primer pairs: Ppdpf_fwd 5'-gtcttcccaaagccgaccc-3' and Ppdpf_rev 5'-gggtggctgctgtgagttta-3', Pdx1_fwd 5'-gtcttcccaaagccgaccc-3' and Pdx1_rev 5'-caaacagcctaaagacaaggct-3'. Ppdpf and Pdx1 were amplified from cDNA derived from (E11) total mouse RNA (Takara) and adult mouse pancreas of C57BL/6J mouse strain, respectively. Purified PCR products were cloned into pCR 2.1 TOPO plasmid (Thermo) according to manufacturer's recommendations with insert:vector ratios of 2:1 and 5:1 with 50 ng vector overnight (o/n) at 16°C. Insert sequences were verified by sanger sequencing (Eurofins Genomics). Plasmid DNA was linearized using HindIII (NEB). To generate DIG-labeled riboprobes, *in vitro* transcription using T7 RNA polymerases (Roche) was performed. Pdx1 antisense [58] and Ppdpf sense probes served as positive and negative controls, respectively. For *in situ* hybridization, mouse tissue was fixed in 4% PFA, cryoprotected in 20% sucrose, and embedded in OCT compound. 18 μm thin frozen sections through the pancreatic anlage of E14.5 C57BL/6J embryos were collected on SuperFrost Plus slides (Thermo). *In situ* hybridization was performed according to [59] with minor modifications. Briefly, after proteinase K treatment and acetylation, sections were permeabilized using 1% TritonX100 and incubated o/n at 63°C with riboprobes diluted

in hybridization buffer [50% formamide, 5x SSC, 5x Denhardt's solution, 250 μg/mL yeast RNA, 500 μg/mL herring sperm DNA]. Sections were washed three times in 0.2x SSC at 65°C, blocked with 10% goat serum, and incubated with alkaline phosphatase conjugated anti-DIG antibody (Roche) at 4° o/n. Alkaline phosphatase activity was detected using nitro blue tetrazolium (NBT) and 5-bromo-4-chloro-3-indolyl-phosphate (BCIP) (Roche).

## Embryonic stem cells

**Maintenance culture.** In this study, the human embryonic stem cell line HUES8 (Harvard University; RRID:CVCL_B207) was used. Culture and differentiation towards the pancreatic lineage was performed with permission from the Robert Koch Institute according to the "79. Genehmigung nach dem Stammzellgesetz, AZ 3.04.02/0084". HUES8 cell authentication was confirmed with a DNA profile using nonaplex PCR of Short Tandem Repeats (STRs) done by the Leibniz-Institute.

Culture conditions for hESCs were 5% $CO_2$, 5% $O_2$, and 37°C. Cells were cultured on hESC Matrigel (Corning) coated plates (according to manufacturer's recommendations) in FTDA medium [60] or mTesR1 medium (STEMCELL Technologies) with daily media change. Splitting in a 1:4 to 1:6 ratio was done twice a week with TrypLE Express (Thermo) to enable feeder-free homogenous monolayer cultures. For this, cells were washed once with PBS and incubated with TrypLE for 3-5 min at 37°C. By adding DMEM-F12+Glutamax (Gibco) the enzymatic reaction was stopped and after centrifugation at 200 x g for 5 min, cells were resuspended in FTDA or mTesR1 supplemented with 10 μM ROCK inhibitor (Y-27632, Abcam) and seeded on a new, coated well.

**Pancreatic progenitor differentiation for *in vitro* experiments.** Differentiation of hESCs into PPs was performed as described in [14]. Briefly, hESCs were seeded on 24-well plates, which were coated with growth factor reduced Matrigel (Corning, diluted 1:18 in DMEM-F12+Glutamax), 24 h prior to the start of differentiation. Cells were cultured in basal media BE1 at d0-d5: MCDB131 (Thermo) with 2 mM L-Glutamine (Gibco), 1.174 g/l sodium bicarbonate (Sigma), 0.8 g/l cell culture tested glucose (Sigma) and 0.1% fatty acid free BSA (Sigma/Proliant). At d6-14 BE3 was used as basal medium: MCDB131 with 2 mM L-Glutamine, 1.754 g/l sodium bicarbonate, 0.44 g/l glucose, 0.5% ITS-X (Gibco), 44 mg/l L-Ascorbic acid (Sigma) and 2% fatty acid free BSA. At a confluency of about 70-95%, differentiation was initialized. For that, cells were washed with PBS (Gibco), and BE1 medium containing 3 μM CHIR99021 (Axon Medchem) and 100 ng/ml ActivinA (Peprotech/R&D) (d0 medium) was added. After initializing differentiation, cells were cultured at 5% $CO_2$, 21% $O_2$, and at 37°C. After 24 h, medium was replaced by BE1 medium supplemented with 100 ng/ml ActivinA (d1 - d3 medium). During the next two days d1 – d3 medium was added to allow formation of definitive endoderm (DE), and on day 4 and day 5 BE1 medium containing 50 ng/ml KGF (Peprotech) was added inducing foregut specification. From day 6 until day 9, cells were cultured in BE3 medium supplemented with 0.25 μM SANT1 (Sigma), 2 μM retinoic acid (Sigma), 0.2 μM LDN193189 (Sigma), and 0.5 μM PD0325901 (Calbiochem). From day 10 on cells received 50 ng/ml FGF10 (Peprotech/R&D), 0.33 μM indolactamV (Stemcell Technologies), 10 μM SB431542 (Axon Medchem), and 16 mM glucose (resulting in a final glucose concentration of 24 mM) in BE3 medium for four days.

**Pancreatic progenitor differentiation for *in vivo* experiments.** For orthotopic transplantation, our most recent protocol was implemented as detailed in [34]. This refined protocol allows higher PP efficiencies and better representation of the multipotency marker GP2. For PE transplantations, planar cultures were harvested at d7 2 days prior to PE and transferred to Costar ultra-low attachment plates (Corning) at 37°C 5% $CO_2$ without orbital shaking. Suspension culture spheres were harvested at the day of transplantation and cells

corresponding to $1 \times 10^6$ PE cells were transplanted. For PP transplants, cells were GP2 MACS sorted at d12 as optimized in [34] (based on [61,62]) and cultured in ultra-low attachment plates at 37°C 5% $CO_2$ without orbital shaking for 1 day. While $0.56 \times 10^6$ sorted PP cells were transplanted in most cases, $10 \times 10^6$ unsorted planar PP cells were transplanted when sorted PP cells were not sufficient for 3 transplantation experiments. In summary, following cells have been transplanted:

(1)  $1 \times 10^6$ suspension PE: 3x PPDPF$^{WT/WT}$, 3x PPDPF$^{KO/KO}$ cl.1, 3x PPDPF$^{KO/KO}$ KO cl.2 (all engrafted well).

(2A)  $0.56 \times 10^6$ GP2-sorted suspension PP: 2x PPDPF$^{WT/WT}$ (1 not engrafted), 2x PPDPF$^{KO/KO}$ cl.1, 3x PPDPF$^{KO/KO}$ KO cl.2

(2B)  $10 \times 10^6$ unsorted planar PP: 1x PPDPF$^{WT/WT}$, 2x PPDPF$^{KO/KO}$ cl.1 (1 teratom, excluded from analysis).

Based on our experience, we suggest transplanting $1 \times 10^6$ unsorted PE or PP cells that were cultured as spheroids at least 1 day prior to transplantation for future experiments.

***In vitro* differentiation into pancreatic endocrine organoids.** We performed stem cell-derived islet differentiation as recently detailed in [36].

## Genome editing by CRISPR/Cas9 in hESCs

**Construct generation.** A large deletion in the *PPDPF* gene locus was generated by CRISPR/Cas9 gene editing using a four times nicking strategy, thereby inducing two DSBs. Two nicks at each intended cleavage site were required to lead to one DSB, thereby minimizing off-target cleavage. crRNAs were designed in intronic regions flanking exon 3 of the *PPDPF* gene locus: guide 1a: GCTGGTTCCGCAGCCTGCAC; guide 1b: GCCTGAAGGCCGGTGGGCTG; guide 2a: GGGGCACTCGGTACTGCTGC; and guide 2b: GAAGCCATTCCCCACCCCCC. CRISPR plasmids were based on pX335-U6-Chimeric_BB-CBh-hSpCAS9n(D10A)-2A-GFP vector (a gift from Boris Greber at MPI Münster [63], itself based on Addgene #42335, a gift from Feng Zhang [64]). Phosphorylation for subsequent ligation of complementary oligonulceotides was performed using the T4 Polynucleotide Kinase (NEB). Oligonucleotides were annealed at 37°C for 30 min and 95°C for 5 min with subsequent cooling to 25°C at 0.1°C/sec. Restriction digestion of the vector was performed using FastDigest BbsI (Thermo, FD1014) for 30 min at 37°C. After digestion, the vector was purified using the JetQuick Genomic DNA Clean up Spin Kit (Genomed) according to the manufacturer's protocol. Ligation of the annealed oligonucleotides with the BbsI-digested vector was performed with a T7 DNA Ligase (NEB) for 1h at room temperature (RT). To prevent undirected recombination, ligation products were treated with a Plasmid Safe ATP-dependent DNase (Biozym, 161010). Electrocompetent DH5A E. coli cells were transformed with final constructs for plasmid amplification. The final DNA of amplified plasmids was checked by PCR and Sanger sequencing.

**Cell line generation.** Before transfection, HUES8 cells were dissociated into single cells using Accutase and 200,000 cells were seeded per well of a 6-well plate. Cells were cultured o/n in FTDA media supplemented with 10 μM ROCK inhibitor. 2 μg DNA (consisting of 0.5 μg of each construct) was transfected via FuGene HD Transfection Reagent (Promega) in OptiMEM medium (Thermo) to one well of 6-well plate for a 4h incubation. After 1 day, transfected hESCs were selected with 1 μg/mL puromycin for 1 day. Multiple single cell-derived colonies were picked mechanically as soon as small colonies had formed for screening of gene edited clones.

**Screening of edited clones: DNA isolation and PCR reaction.** Clonal colonies were mechanically dissociated and one half of the cells was further cultivated and expanded while the other half was used for genotyping. DNA was isolated using the QIAamp DNA Micro Kit (Qiagen), according to manufacturer's instructions and amplified using a high-fidelity PrimeSTAR GXL DNA Polymerase (Takara). PCR screening was based on a primer pair spanning the target region (fwd: GAGCCCCGCCTCGGTAAATAAC; rev: AGGGTGGACTTCCCGAAAAAGAA). PCR products were sent for Sanger sequencing (GATC Biotech AG Tuebingen) for validation of clonal genotypes (sequencing primer: CTCGGTAAATAACCCAGC). For off-target sequencing, following primer pairs were used: For MGMT, fwd: TCAGGAGCACCCATTTGGTC, and rev: GGCCCCCATTTAAGGGTTCA and for TTC34, fwd: AAACGCACCCACAGACCG, and rev: TGAGGGTGGGGTTTCTGTTC.

## Sample processing and molecular biology assays

**Co-Immunoprecipitation experiments.**
<u>Construct generation</u>
While expression plasmids of GATA4, FOXA1, FOXA2, PDX1, NKX6-1 with C-terminal Flag- and GFP-tag have been previously generated in a pcDNA3 backbone [13], PPDPF, KCTD12, and MAGED1 expression plasmids with the same tags were cloned in this study. PPDPF constructs were commercially synthesized (Eurofins) into a pEX-A backbone and recloned into our library of pcDNA3, pcDNA3-Flag, and pcDNA3-GFP via EcoRI and NotI restriction enzymes. For MAGED1 and KCTD12, the coding sequence (CDS) was amplified from human cDNA and first TOPO-cloned into pCR XL TOPO and pCR 2.1 TOPO, respectively, using TA cloning kits (Thermo). During PCR amplification, enzyme recognition sites were introduced with following primers and GoTaq Flexi Polymerase (Promega): MAGED1 (NotI fwd: ctgaacGCG GCCGCaaGCTCAGAAAATGGACTGTGGTG; XbaI rev: ctgcacTCTAGAtgcTCACTCAAC CCAGAAGAAACCAA); KCTD12 (EcoRI fwd: ctgaacGAATTCGCTCTGGCGGACAGCAC; Xba rev: ctgcacTCTAGATCACTCCCTGCAGAAGACGTAC). For KCTD12 PCR reaction, 3% DMSO was added. Topo cloning was performed according to manufacturer's recommendation with insert:vector ratios of 2:1 and 5:1 with 50 ng vector and o/n TA reaction at 16°C. For transformation and bacteria amplification, chemical competent DH5a were used for PPDPF and MAGED1, while dam incompetent chemical competent bacteria (JM110) were employed for KCTD12 due to a Dam methylation blocked XbaI site. After topo cloning and validation by sanger sequencing (Eurofins Genomics), CDSs of interest were cloned into target pcDNA3-Flag and pcDNA-GFP vectors via restriction enzyme digestion. Sticky ends were produced for MAGED1 plasmids with NotI-XbaI, for PPDPF and KCTD12 with EcoRI-XbaI and cloned via T4 DNA ligation (NEB) using insert:vector ratios of 3:1 with 100 ng vector input.
<u>Lipofection and protein lysis of HEK293T cells</u>
2.1 x 10^6 HEK293T cells were seeded in DMEM with 10% FCS and 1x Penicillin/Streptomycin on a 10 cm dish 1 day prior to transfection. For transfection, 10 μg plasmid DNA was transfected with 18 μL Lipofectamine 2000 (Thermo). After 24 h, cells were washed with PBS and harvested with 0.25% Trypsin/EDTA. After one additional PBS washing step, cells were resuspended in 600 μL chaps lysis buffer (0.01 M Chaps + 50 mM Tris-HCL (pH 7.8) + 150 mM NaCl) supplemented with 5 mM NaF + 0.5 mM PMSF + 1x protease inhibitor cocktail (Sigma) and lysed on ice for 1 h.
<u>Immunoprecipitation experiments</u>
After washing agarose-conjugated α-Flag M2 beads (Sigma) 6 times with 0.5% BSA in PBS, beads were centrifuged at 420 x g for 1 min at 4°C and protein extracts were added to 40 μL

beads and put on a shaker o/n at 4°C. After incubation, samples were washed 6 times with chaps buffer and resuspended after centrifugation at 20,800 x g for 1 min at 4°C in 1x laemmli buffer. Proteins were detected via Western Blotting as detailed below.

**Western Blotting.** For protein extraction of hESCs and differentiated pancreatic cells, cell lysates were generated by incubating cell pellets in RIPA buffer supplemented with 1 mM PMSF, phosphatase inhibitor (1x) and protease inhibitor cocktail (1x) for 30 min on ice and vortexing every 10 min. After 8 min centrifugation at 10,000 x g, supernatant containing the protein fraction was collected. Protein concentration was determined using a Bradford reagent (Bio Rad) and equalized amounts of protein lysates were separated on a 10-12% polyacrylamide gel in SDS-buffer followed by blotting to a methanol-activated Immobilon-P PVDF membrane (Millipore) by using semidry transfer buffer (32 mM glycine (Applichem), 44 mM Tris, and 20% methanol (Sigma)) and the Transblot semidry transfer system (Bio-Rad). Effective protein transfer was confirmed by Ponceau staining (AppliChem) before membrane was blocked with 5% BSA and 0.1% Tween20 (Sigma) in TBS (TBS-T) for at least 1 h at RT. Membranes were incubated with primary antibodies diluted in blocking solution o/n (o/n) at 4°C. After washing three times with TBS-T, incubation with secondary antibody anti-mouse-horseradish peroxidase (HRP) or anti-rabbit-HRP (ECL anti-rabbit or mouse IgG, GE Healthcare) was performed for 1 h at RT. HRP was detected with the SuperSignal West Dura kit (Thermo) on a chemiluminescence imaging fusion SL system (VILBER). The following primary antibodies were used: anti-GFP (Roche, 1181 446 0001, 1:500), anti-Flag M2 (Sigma, F3165, 1:500), anti-PDX1 (R&D, AF2419, 1:500), anti-PPDPF (Atlas Antibodies, HPA040929, 1:150), and anti-Vinculin (Sigma, V9264, 1:1000).

**Flow cytometry of surface marker.** At PSC stage, pluripotency marker SSEA4 and TRA1-60 were checked after gene editing. At DE stage, differentiation efficiency was determined by KIT (CD117) and CXCR4 (CD184) surface marker staining. For each measurement, cells from one well of a 24-well plate were harvested with TrypLE, enzymatic reaction was stopped with FC buffer containing 2% FCS in PBS and washed once with FC buffer (200 x g, 5 min). Cells were blocked for at least 20 min on ice with blocking buffer consisting of 10% FCS in PBS and washed again with FC buffer. After resuspension of the cell pellets in 50 μl FC buffer, DE cells were incubated with PE-conjugated CXCR4 antibody (Thermo, MHCXCR404, 1:50) and APC-conjugated KIT antibody (Thermo, CD11705, 1:100) for 45 min on ice. hESCs were incubated with PE-conjugated SSEA4 (BD Bioscience, 560128, 1:10) and FITC-conjugated TRA1-60 (BD Bioscience, 560380, 1:10) for 60 min on ice. Samples were washed and resuspended in FC buffer and filtered using a 50 μm polyamide mesh (Hartenstein). DAPI was added after the last washing step at a concentration of 150 ng/μl to distinguish viable (DAPI$^-$) and dead (DAPI$^+$) cells during analysis.

**Flow Cytometry of Annexin V.** For Annexin V staining, cells were harvested as described for surface marker staining and staining was performed according to manufacturer's instructions of the FITC Annexin V Apoptosis Detection Kit with 7-AAD (Biolegend, 640922). Instead of using the recommended volumes of reagents, applied volumes were halved without changing reaction mix concentrations.

**Flow cytometry of intracellular marker.** Differentiation efficiency was analyzed at PE and PP stage by FC-based analysis of intracellular PDX1 and NKX6-1 expression. Endocrine organoids were singularized with TrypLE for around 9-12 min with pipetting up and down in-between and control of proper singularization under the microscope. Planar cells were harvested and singularized as for surface marker staining. Singularized cells were washed with PBS (200 x g, 5 min) and fixed on ice for 25 min in 4% PFA in PBS with 100 mM sucrose (Sigma). Samples were washed twice with PBS and blocked with 5% donkey serum (Jackson ImmunoResearch) in 0.1% Triton-X/PBS for 30 min on ice. After centrifugation (3000 x

g, 5 min) and two rounds of washing with 2% donkey serum, 0.1% Triton-X in PBS (wash solution), cells were resuspended in blocking solution with primary antibodies. Incubation was performed o/n at 4°C or 90 min on ice with combination of following antibodies: anti-PDX1 (R&D, AF2419, 1:500), anti-NKX6-1 (DSHB, F55A12, 1:150), anti-Ki67 (Thermo, MA5-14520, 1:1000 or DAKO, M7240, 1:200 or BV421-coupled (BD Horizon, 562899, 1:20)), anti cl-CASP3 (Cell Signaling, 9661, 1:800), GCG (Sigma, G2654, 1:500), C-pep (Cell Signaling, 4593, 1:100), and INS-Alexa647 (Cell Signaling, 3014, 1:80). Next, the samples were washed twice and incubated with Alexa Fluor secondary antibodies (Thermo, 1:500) for 90 min on ice. After one additional washing step, cells were finally resuspended in washing solution and filtered to obtain single cells before measurement. FC measurements were performed on an LSR II (BD) or an AttuneNxT (Thermo) flow cytometer.

**Flow cytometry of EdU staining.** EdU staining was performed using the Click-iT EdU Alexa Fluor 647 Assay Kit (Thermo, C10635). EdU at a concentration of 10 μM was added to the cell cultures prior to harvesting. Chosen incubation time was optimized for each cell type: At hPSC stage, EdU was incubated for 45 min, at DE stage for 2 h and at PE and PP stage for 4 h. Cells were harvested, fixated as described in the intracellular flow cytometry section and staining was performed according to manufacturer's recommendations with minor changes. Instead of using the recommended volumes of reagents, applied volumes were halved without changing reaction mix concentrations. DNA was finally incubated with 1 μg/μL DAPI for around 30min prior to measurements on an AttuneNxT (Thermo) flow cytometer.

**Flow cytometry analysis.** Analysis of flow cytometry data including gating was performed using FLowJo version 10.8.1. Briefly, Cells were gated on the area of forward and sideward scatters. Single cells were subsequently defined by the area and width of the forward scatter followed by an additional stringency step using the area and width of the sideward scatter. In case of live cell measurements, DAPI$^-$ or 7AAD$^-$ cells were gated as living cells for subsequent marker analysis (area of specific channels).

**Immunocytochemistry staining.** Prior to immunocytochemistry (ICC/IF) in-well staining, stem cells were cultivated and differentiated on Ibidi-precoated glass-bottom IBIDI-24-well plates (IBDI) with additional Matrigel coating as outlined in maintenance and differentiation culture. Cells were washed with PBS, fixed in 4% PFA+100 mM sucrose solution at RT for 20 min, and washed with PBS three times. Quenching was performed with 50 mM NH$_4$Cl (Sigma) for 10 min and wells were washed three times with PBS before permeabilization with 0.1% Triton-X/PBS was performed for 30 min at 4°C. Cells were blocked with 5% normal goat or donkey serum (Jackson ImmunoResearch) in 0.1% Triton-X/PBS for 45 min and incubated with the primary antibody solution at 4°C o/n. The following antibodies were used: anti-OCT4 (Santa Cruz, sc-5279, 1:100), anti-SOX2 (R&D, MAB2018, 1:300), anti-SOX17 (R&D, AF1924, 1:500), anti-PDX1 (R&D, AF2419, 1:300), and anti-NKX6-1 (DSHB, F55A12 (concentrate), 1:150). On the next day, cells were washed three times with PBS and incubated with secondary Alexa Fluor antibodies (Thermo) 1:500 diluted in blocking solution at RT for 1 h in the dark. After washing with PBS, 500 ng/ml DAPI in PBS was added to the cells for 10 min. Wells were either directly imaged or stored in PBS at 4°C prior to imaging. Cells on IBIDI-plates were imaged on an AxioObserver 7 microscope (Zeiss) equipped with a Axiocam 702 camera (Zeiss) using ZEN blue software (Zeiss).

## Histology and immunohistochemical staining

### Histological processing
Pancreata of transplanted mice and endocrine organoids were fixed with 4% paraformaldehyde solution (PFA) at 4°C o/n. Dehydration, paraffin embedding, and sectioning (4 μm)

was performed according to standard histology procedures. Hematoxylin and eosin (H&E) staining was applied to identify human engraftments.

Immunofluorescence (IF)

Rehydration in an ethanol series was followed by heat-mediated antigen retrieval using either tris buffer (pH 9) or citrate buffer (pH 6, both Vector Laboratories) for 20-25 min in a steamer. For IF, tissue permeabilization was performed with 0.5% Triton X-100/PBS (PBS-T) for 30 min at RT. After washing twice, primary antibodies diluted in Antibody Diluent (Zytomed) were added to the slides, which were then incubated o/n at 4°C in a wet chamber. After washing three times with PBS-T for 5 min, pancreatic tissue slides were quenched with TrueView (Vector Laboratories) for 5 min at RT. No quenching was performed for endocrine organoid sections. Next, slides were stained with Alexa Fluor secondary antibodies (Thermo) and 1 µg/ml DAPI diluted in Antibody Diluent for 90 min at RT in the dark. Slides where washed three times with PBS-T and finally with dH2O before sections were mounted with Vectashield Vibrance Antifade mounting medium (TrueView kit, different mounting medium might affect quenching). Following ABs were employed: AcTUB (Abcam, ab179484, rabbit, 1:1000, steamer citrate), CHGA (DAKO, M0869, mouse, 1:200, steamer citrate), KRT19 (DAKO, M0888, mouse 1:100, steamer tris), CPA1 (BioRad. AHP2054, rabbit, 1:1000, steamer tris), TRYP (Santa Cruz, sc-137077, mouse, 1:50, steamer tris), AFP (DAKO, A0008, rabbit, 1:500, steamer citrate), and CTRC (Millipore, MAB1476, mouse, 1:500, steamer citrate). For endocrine organoids, following ABs were used: PDX1 (R&D, AF2419, 1:500), NKX6-1 (DSHB, F55A12 (concentrate), 1:150), GCG (Sigma, G2654, 1:1000), C-pep (Cell Signaling, 4593, 1:200), and INS (Abcam, ab181547, 1:5000). IF images were acquired on an Axio Observer 7 microscope (Zeiss) equipped with a Axiocam 702 camera (Zeiss) using ZEN blue software (Zeiss).

Immunohistochemistry (IHC)

For immunohistochemistry (IHC) staining, antigen retrieval was directly followed by incubation with primary antibodies diluted in Antibody Diluent (Zytomed) for 30 min at RT in a wet chamber. Slides were washed and antibody detection was performed applying alkaline phosphatase-based Dako real Detection Kit (Agilent). In brief, slides were incubated first with a biotinylated secondary antibody, washed and next incubated with Streptavidin both for 30 min at RT. After another washing step, slides were incubated with Red Detection Dye for around 10 min, counterstained with 20% Hematoxylin solution (Merck) for 30 sec, washed with tap water for 5 min and finally mounted with Aquatex (Merck). For IHC of PDX1 raised in goat, a biotinylated secondary antibody against goat IgG (Vector Laboratories) was used in combination with Vectastain Elite ABC-HRP kit and Novared peroxidase substrate kit (both Vector Laboratories). Following Abs were employed: NKX6.1 (DSHB Hybridoma, F55A12, mouse, 1:150, steamer tris), CDX2 (Cell Marque, 235R-14, rabbit, 1:500, steamer tris), PDX1 (PDX1 (R&D, AF2419, goat, 250, steamer tris), C-pep (Cell Signaling, 4593, rabbit, 1:200, steamer citrate), GCG (Sigma, G2654, mouse, 1:1000, steamer citrate), P53 (Santa Cruz, sc-47698, mouse, 1:100, steamer citrate), P21 (Abcam, ab109520, rabbit, 1:300, steamer citrate), Ki-67 (DAKO, M7240, mouse, 1:500, pressure cooker), Human Nucleoli antibody (Abcam, ab190710, mouse, 1:200, pressure cooker). IHC images were acquired on a DM5500B microscope equipped with a DMC5400 camera (Leica) using the Application Suite software (Leica).

**ICC, IF, and IHC image analysis.** For ICC/IF and IF, brightness and contrast was adjusted in ZEN blue (Zeiss), for IHC in Photoshop (Adobe) according to good scientific practice. Images were then exported as jpeg files and cropped in Adobe Illustrator (Adobe) for final image compilation.

**Analysis of the PPDPF interactome.** The SILAC-based analysis of the PPDPF interactome was performed as previously described (PMID: 250s59612). Briefly,

HEK293T cells were metabolically labeled with lysine and arginine containing light (K0/R0) (Sigma; control cells) or heavy (K8/R10) carbon, nitrogen or hydrogen isotopes (Silantes) for ten passages to allow complete labeling of proteins. Labeled cells were transfected with N-terminally Strep/FLAG-tagged PPDPF or a vector control by lipophilic transfection via a home-made PEI reagent. To rule out labeling artifacts, labels were also switched between the conditions. The bait protein and associated proteins were purified via StrepTactin resin (IBA) and eluted with Desthiobiotin elution buffer (IBA). The eluates of the heavy and light condition were mixed 1:1 and subjected to mass spectrometric analysis. For this purpose, samples were precipitated by Methanol/Chloroform and cyteines were alcylated by DTT/Iodoacetamide prior to a tryptic proteolysis following standard conditions. After the proteolysis step, samples were desalted by StageTips (Thermo-Fisher) and analysed on a Q-Exactive Orbitrap mass spectrometer (Thermo-Fisher) coupled to a nano-flow high-performance liquid chromatography (HPLC) system (Dionex Ultimate 3000 RSLC, Thermo-Fisher). For separation, 180 min standard gradients were used. To obtain MSMS spectra, the ten most intense precursor peptides were subjected to fragmentation (TOP10 method). Resulting RAW files were analysed by MaxQuant (ver. 1.6.2.10) setting K8/R10 as heavy and K0/R0 as light condition. The search was performed against the human subset of the Swissport/UniprotKB database (release: 2016_09, 23442 entries) using carbamylated cysteine as fixed and methionine oxidation and N-terminal acetylation as variable modifications. The statistical analysis was performed with Perseus (ver. 1.6.7.0).

**RNA isolation, reverse transcription and qPCR.** Cells were harvested with TrypLE as outlined above. RNA was extracted from cell pellets with the GeneJET RNA Purification Kit (Thermo) according to manufacturer's instructions. Reverse transcription of 200-1000 ng of total RNA was performed with the iScript cDNA Synthesis Kit (Bio-Rad) and cDNA was utilized for quantitative real-time PCR (qPCR) with SensiMix SYBR No-ROX Kit (Bioline) on the QuantStudio 3 Real-Time PCR System (Thermo). Following commercial QuantiTect primers (Qiagen) of target genes were employed: Hs_SOX17_1_SG (QT00204099), Hs_CXCR4_2_SG (QT02311841), Hs_GATA6_1_SG (QT00233331), Hs_GATA4_1_SG (QT00031997), Hs_FOXA1_1_SG (QT00212828), Hs_SOX9_1_SG (QT00001498), Hs_PDX1_1_SG (QT00201859), Hs_PROX1_1_SG (QT01006670), Hs_NKX6-1_1_SG (QT00092379), Hs_PTF1A_2_SG (QT01033396), Hs_GP2_1_SG (QT00010535), Hs_CPA1_1_SG (QT00001736), Hs_PPDPF_1_SG (QT00203469). Target genes were normalized against the housekeeping gene Hs_HMBS_1_SG (QT00014462) using $2^{-\Delta\Delta Ct}$.

## RNA-seq experiments

**RNA isolation, library preparation and sequencing.** RNA from hPSC, DE, PE and PP stage was isolated as described above. For DE, PE, and PP, RNA was isolated from four independent differentiations (differentiation started on different days). For hPSC control samples, 2 RNA samples were isolated. Quality and concentration were measured on an RNA Pico chip on an Agilent Bioanalyzer. Library preparation and sequencing was performed as detailed in [65]. Briefly, the Sequencing library preparation has been performed with the TrueSeq RNA Sample Prep Kit v2-Set B (RS-122–2002, Illumina Inc, San Diego, CA) producing 275 bp fragments including adapters in average size. Pooled libraries have then been clustered on the cBot Instrument from Illumina using the TruSeq SR Cluster Kit v3—cBot—HS(GD-401–3001, Illumina Inc, San Diego, CA). Sequencing was performed as 50 bp single reads and 7 bases index reads on an Illumina HiSeq2000 instrument using the TruSeq SBS Kit HSv3 (50-cycle) (FC-401–3002, Illumina Inc, San Diego, CA).

**Data processing.** RNA-seq reads were aligned to the human genome using STAR Aligner v.2.5.2a [66] with the Ensembl 86 reference genome. Sequenced read quality was checked with FastQC v0.11.2 (http://www.bioinformatics.babraham.ac.uk/projects/fastqc/) and alignment quality metrics were calculated using the RNASeQC v1.18 [67]. Following read alignment, duplication rates of the RNA-seq samples were computed with bamUtil v1.0.11 to mark duplicate reads and the dupRadar v1.4 Bioconductor R package for assessment [68]. Unique read counts were retrieved from the "featureCounts" software package [69]. As quality control measure, Reads Per Kilobase of transcript per Million mapped reads (RPKM) were quantified using Cufflinks software version 2.2.1 [70].

For quality control, total read counts, the rate of ribosomal RNA, the rate of intergenic and intronic reads and distribution of logarithmic RPKM values were assessed for each sample. While all samples except of one were of very good quality with 20-40 million 85 bp single end reads and very low ribosomal, intergenic, and intronic rates, one sample (PPDPF$^{WT/WT}$ cl. 1 at DE, replicate 3) was excluded from downstream analysis. Following samples were finally analyzed:

hESC/hPSC (day 0): 4 genotypes x 2 replicates = 8 samples

DE (day 4): 4 genotypes x 4 replicates - 1 sample exclusion = 15 samples

PE (day 10): 4 genotypes x 4 replicates = 16 samples

PP (day 14): 4 genotypes x 4 replicates = 16 samples

**RNA-seq data analysis.** Generation of DEG lists and transformed read counts
Raw read counts were imported to R v4.2.1 and differential expression was analyzed with "DESeq2" [71]. A gene was determined to be differentially expressed if the absolute log2 FC > I1I and the adjusted P-value < 0.01. Read counts were Rlog transformed, before subjected to downstream statistical analysis.

PCA
Prior PCA analysis, Rlog transformed read counts were filtered for the top 10% most variant genes. For the PE and PP subset PCA, Rlog transformed read counts were first filtered for PE and PP samples before filtering for the top 10% most variant genes. Subsequently, PCA analysis was performed using the "prcomp" function and standard settings in R. PCA was visualized using "ggbiplot" function in conform package.

Hierarchical clustering
Sample distances have been calculated from Rlog transformed read counts using the "sample-Dists" function with default parameters such as Euclidean distance calculation.

Overrepresentation analysis
Overrepresentation analysis was conducted on pairwise DEG lists using the web browser tool of gProfiler [56].

Venn diagrams and Volcano plots
Venn diagrams were generated using "ggvenn" function and package, while "ggplot" was used for plotting Volcano plots.

Line plots and heatmaps
Line plots and heatmaps illustrating target gene expression were performed on Rlog transformed read counts. The Mean ± SEM was plotted with "ggplot" along the time course of differentiation for line plots. Heatmaps were generated using the packages ("pheatmap") for plotting and the package ("biomaRt") for retrieval of gene sets from common databases. The function ("pheatmap") was used with parameter scale="row" and clustering_method="ward.D2" with further parameters set as default.

Gene set enrichment analysis (GSEA)

GSEA was conducted with the desktop version of GSEA version 4.0.3 (Broad Institute) [72]. Run GSEA function (not preranked function) was conducted on the Rlog transformed read count matrix and an additionally prepared data information file. Except of Collapse parameters ("No_collaps"), default settings including 1000 permutations and a weighted statistical analysis were used. For comparison with reference gene sets, gene lists have been either directly compiled from literature or raw data has been reanalyzed. A complete list of applied gene sets can be found in S5 Table.

Generation of reference gene list

Differentiation: For stage-specific gene sets of hPSC-based pancreatic differentiations, two different RNA-seq data sets were utilized. The gene sets for Definitive endoderm_Xie2013, Gute tube_Xie2013, Pancreatic endoderm_Xie2013, Pancreatic progenitor_Xie2013 were directly retrieved from [27]. For Pluripotent stem cell_Breunig2021 upregulated DEGs of the pairwise RNA-seq comparison between hPSC and DE from [23] were extracted, for Definitive endoderm_Breunig2021 the downregulated DEGs of the same comparison and for Pancreatic progenitor_Breunig2021 upregulated DEGs of the comparison PPs versus DE were listed.

Mouse Ptf1a Targets_Thompson2012: Gene sets from an RNA microarray of Ptf1a$^{KO/KO}$ versus Ptf1a$^{KO/WT}$ murine pancreatic progenitors could be directly retrieved from [31] (Ptf1a$^{KO/KO}$ up_Thompson2012 and Ptf1a $^{KO/KO}$ down_Thompson2012).

Human Fetal Tissue_Jennings2017: A human fetal gene set was generated from raw FASTQ files (friendly provided by Neil Hanley) [29]. Briefly, FASTQ files have been mapped to the human genome, normalized using batch correction, and significant gene lists have been generated by Limma pairwise comparisons between all investigated organs. All genes found significant in at least on pairwise comparison were subsequently sorted according to their peaked expression to assign organ-specific gene lists (Liver, Extrahepatic biliary duct, Pancreas).

Human Fetal Pancreas_Gonçalves2021: Gene lists were directly retrieved from cluster analysis in supplementary tables of [73] (Unknown1_Gonçalves2021, Erythroblasts_Gonçalves2021, Proliferative_Gonçalves2021, Mesenchyme_Gonçalves2021, Tip_Gonçalves2021, Trunk_ Gonçalves2021, Endocrine_Gonçalves2021, Neural_Gonçalves2021).

Human Fetal Tissue_Cao2020: Organ-specific pseudobulk expression profiles were retrieved from S2 Table of [30]. For final gene lists, genes with P-value < 0.01 and log2 FC > I2I were selected (Adrenal_Cao2020, Placenta_Cao2020, Cerebellum_Cao2020, Lungs_Cao2020, Liver_Cao2020Stomach_Cao2020, Intestine_Cao2020, Kidney_Cao2020, Brain_Cao2020, Muscle_Cao2020, Eye_Cao2020, Spleen_Cao2020, Heart_Cao2020, Pancreas_Cao2020, Thymus_Cao2020).

**Reanalysis of human fetal and adult pancreas scRNA-seq data.** The processed gene expression matrix and annotation data (including developmental time point, cell type annotation and UMAP coordinates) of the human fetal pancreas dataset [32] were downloaded from OMIX database under accession code OMIX001616 and reanalyzed using seurat v5.1.0 [74]. Counts were normalized using Seurat´s "NormalizeData" function. The expression of selected genes across developmental time points and cell types and within UMAP space was visualized using Seurat´s "VlnPlot" and "FeaturePlot" functions, respectively, and kernel density estimates of gene expression were plotted using "Nebulosa´s plot_density" function.

The adult pancreas scRNA-seq data, downloaded from GSE84133 has been previously reanalyzed as detailed here [23].

**ChIP-seq and ATAC-seq analysis.** ChIP-seq and ATAC bigwig files have been downloaded from publicly available data as described in the data availability section. Data has

been plotted at the PPDPF locus with the desktop version of Integrative Genomics Viewer (IGV) version 2.6.3 [54].

**PheWAS studies.** Association results were extracted from HugeAMP (https://hugeamp.org) [20], cis-eQTLS from (https://eqtlgen.org/cis-eqtls.html) [22], FORGEdb scores from https://forgedb.cancer.gov [55], and SNP frequencies in ethnical cohorts (only listed in S1 Table) from https://www.ncbi.nlm.nih.gov/snp.

**Figure preparation.** Adobe Illustrator was used for compilation and arrangement of multipaneled figures and for the generation of schematic views. The in Figs 9A and S6A depicted scheme of an orthotopic transplantation was produced with the help of the generic artificial intelligence function of Adobe Illustrator.

**Statistical analysis.** If not stated elsewhere, statistical analysis was performed using the GraphPad Prism 8 software and detailed information regarding the different applied tests are indicated in figure legends. In general, data summarize three independent experiments (independently started hPSC differentiations) with each analysis performed in duplicates (two wells per condition), unless otherwise stated. Statistical analyses was performed on the respective means of technical replicates. Statistical significance was defined as follows: * P-value < 0.05, ** P-value < 0.01, *** P-value < 0.001, **** P-value < 0.0001. For differentiation experiments, independently started wells were considered independent experiments, while wells of the same differentiation were treated as technical replicates.

## Supporting Information

**S1 Fig. Regulation of *PPDPF* expression in human *in vitro* pancreatic differentiation. (A)** ATAC-seq (Assay for Transposase-Accessible Chromatin using sequencing)-based chromatin opening at PPDPF locus from our previous *in vitro* differentiation of the hESC line HUES8 [13]. **(B)** Histone marks along the time course of differentiation in CyT49 differentiation complementing main Fig 2F. **(C)** Chromatin immunoprecipitation (ChIP)-seq binding peaks of murine Foxa2, Ptf1a, and Rbpjl from publicly available ChIP-seq data at E17.5 [75]. ATAC- and ChIP-seq peaks have been visualized via Integrative Genomics Viewer (IGV) [54]. (TIF)

**S2 Fig. CRISPR/Cas9 gene editing of *PPDPF* in hPSCs. (A)** Scheme of the applied four times CRISPR/Cas9 nicking strategy to delete exon 3 of *PPDPF*. **(B)** PCR-based clone screening after gene editing in the hESC line HUES8 highlighting the two employed PPDPF$^{KO/KO}$ clones. Note that the PCR band size, which was generated with primers spanning the target region, decreases after successful gene editing (from around 550 bp to around 400 bp). **(C)** Sanger sequencing of the two PPDPF$^{KO/KO}$ clones. Both clones harbored identical homozygous mutations. **(D)** Predicted alterations on protein level caused by the induced genetic deletion. **(E)** *PPDPF* mRNA expression relative to *HMBS* in PPDPF$^{KO/KO}$ and PPDPF$^{WT/WT}$ PPs (n = 3; Mean ± SEM; ordinary one-way ANOVA followed by Dunett's multiple comparison test; cl.: clone). **(F)** Western Blot at PE stage confirming KO of PPDPF on protein level (200 µg, n = 2). **(G)** Flow cytometry analysis of pluripotency marker SSEA4 and TRA1-60 in PPDPF$^{KO/KO}$ and PPDPF$^{WT/WT}$ hESCs (n = 1, in duplicates; Mean ± SD). **(H)** ICC/IF staining of pluripotency marker OCT4 and SOX2 in PPDPF$^{KO/KO}$ and PPDPF$^{WT/WT}$ hPSCs (scale bar: 100 µm, n = 1). (TIF)

**S3 Fig. Off-target analysis in gene edited clones. (A)** Overlap between CRISPOR [24] predicted potential off-targets of different guides. Two nicks on complementary strands and in close proximity are required for induction of a double strand break (DSB). **(B)** Overlap between predicted potential off-targets and RNA-seq DEGs. Only 2 genes were found, of

which expression was significantly reduced in at least 3 stages. **(C)** Line plots depicting Mean ± SEM of the two genes, *MGMT* and *TTC34*, with **(D)** subsequent sanger sequencing on the potential off-target sites. Sequences were unaltered in all clones. One representative PPDPF[WT/WT] and PPDPF[KO/KO] hESC clone sequence is shown. **(E)** Exemplary line plots of genes (Mean ± SEM), in which potential off-targets were shared between guide1a and 1b, respectively 2a and 2b. Also, other pairwise comparisons (e.g. guide 1a with 2a) were investigated with no indications for a loss- or gain-of-transcript mutation.
(TIF)

**S4 Fig.  Depletion of pancreatic identity in PP differentiation upon PPDPF ablation.**
Gene set enrichment analysis (GSEA) for gene sets compiled from **(A)** an RNA microarray of Ptf1a[KO/KO] versus Ptf1a[KO/WT] murine pancreatic progenitors [31], **(B)** bulk RNA-seq from human fetal pancreas [29], **(C)** scRNA-seq from human fetal pancreas [73] and **(D)** scRNA-seq from human fetal tissues [30]. For all: False discovery rates (FDRs) ≤ 0.25 are highlighted in red (left panel), while comparisons that were enriched in PPDPF[KO/KO] cells are depicted in green and comparisons that were depleted in blue (right panel). GSEA plots against depicted reference gene sets are shown for the comparison of PPDPF[WT/WT] and PPDPF[KO/KO] PPs (only for proliferative progenitor of Gonçalves PE plotted). Line plots (Mean ± SEM) are displayed for genes, top ranked in GSEA. Additional information: For "PTF1A[KO/KO] up[Thompson 2012]", 4 out of 64 genes were significantly upregulated (DEGs) in PPDPF[KO/KO] PPs over PPDPF[WT/WT] PPs, while 0 were downregulated. For "PTF1A[KO/KO] down[Thompson 2012]", 0 out of 66 genes were up, 1 was down. For "liver[Jennings 2017]", 4 out of 79 genes were up, 2 were down. For "Extrahepatic biliary duct[Jennings 2017]", 1 out of 32 genes was up, 5 down. For "Pancreas[Jennings 2017]", 0 out of 32 genes were up, 3 down. For "Proliferative/Progenitor[Gonçalves 2021]", 2 out of 195 genes were up, 1 was down (at PE stage). For "Tip[Gonçalves 2021]", 0 out of 111 genes were up, 1 was down. For "Trunk[Gonçalves 2021]", 0 out of 55 genes were up, 2 were down. For "Placenta Pancreas[Cao 2020]", 20 out of 307 genes were up, 4 were down. For "Intestine Pancreas[Cao 2020]", 7 out of 239 genes were up, 0 down. For "Pancreas[Cao 2020]", 0 out of 107 gene were up, 1 was down.
(TIF)

**S5 Fig.  scRNA-seq analysis of human fetal and adult pancreas.** UMAP representations and violin plots for expression of **(A)** progenitor marker *PTF1A*, **(B)** *PDX1*, **(C)** *FOXA1*, **(D)** *FOXA2*, **(E)** acinar marker *CEL*, **(F)** ductal marker *CFTR*, **(G)** endocrine marker *CHGA*, and **(H)** trunk (and tip) marker *TTYH1* in human fetal pancreas [32] complementing main Fig 8A-D. **(I)** Expression of *PPPDF* and selected cell type marker in human adult pancreas [33] revealing relatively broad expression of *PPDPF*. EP: endocrine progenitor; MP: multipotent progenitor; UMAP: Uniform Manifold Approximation and Projection.
(TIF)

**S6 Fig:  PPDPF[KO/KO] PPs successfully develop into pancreatic tissue upon orthotopic transplantation. (A)** Schematic overview of orthotopic transplantation experiments and flow cytometry-based quality control of GP2 sorting (n = 1). **(B)** Hematoxylin and eosin (H&E) staining of identified grafts with n = 2 for PPDPF[WT/WT] PPs and n = 3 for PPDPF[KO/KO] cl.1 and cl.2 PPs. Unsorted planar PPs were used when cell numbers after GP2 sort were not sufficient. **(C)** Human nucleoli (H-NUCL) staining illustrates human engraftment. **(D)** Quantification of depicted marker expression averaged over individual grafts of each genotype (– completely or nearly completely absent; + weak expression; ++ moderate expression; +++ strong expression. **(E)** Immunofluorescence (IF) staining of acinar marker CPA1, Trypsin, and CTRC, ductal marker CK19, and liver marker AFP. **(F)** IF and immunohistochemistry (IHC) of ductal cilia marker acetylated tubulin (acTUB), endocrine marker CHGA, GCG, and C-peptide (C-pep).

**(G)** IHC of progenitor marker PDX1, NKX6-1, and CDX2. **(H)** IHC of proliferation marker Ki-67 and effectors of growth restriction P53 and P21. For all: Images are depicted that show strong marker expression to highlight trilineage competence. Scale bar: B-C: 500 μm, inlet: 100 μm. E-H: 100μm, inlets 2x enlarged.
(TIF)

**S1 Table. GWAS traits, frequency of SNPs, and information about credible set.**
(XLSX)

**S2 Table. List of PPDPF-bound proteins identified in HEK293T cells via SILAC and gProfiler of PE/PP intersected proteins.**
(XLSX)

**S3 Table. Stage-specific lists of differentially regulated genes (DEGs).**
(XLSX)

**S4 Table. Stage-specific overrepresentation analysis using gProfiler.**
(XLSX)

**S5 Table. Developmental gene sets compiled from literature.**
(XLSX)

**S6 Table. Values behind depicted graphs.**
(XLSX)

## Acknowledgements

The authors thank Katrin Jochmann, Katrin Köhn, Ulrike Mayr-Beyrle, and Ralf Köhntop for excellent technical assistance, and Justin Antony, Michael Kormann (University of Tuebingen), Franz Oswald, Sabine Schirmer (Ulm, University), Ninel Azoitei, Johann Gout, Elodie Roger, Michael Karl Melzer, and Sandra Heller (IMOS) for helpful project discussions. They acknowledge Rachel E. Jennings and Neil A. Hanley (University of Manchester) for providing FASTQ raw data on human fetal RNA-seq and thank Thomas Engleitner, Gaurav Jain, and Roland Rad (Technical University Munich) for their support in reanalysis of this data set and the additional human adult scRNA-seq pancreas data set. They thank Ivan Costa, Nicole Zhang, and Kevin Menden for providing input on RNA-seq data evaluation and O.D. Madsen and the Developmental Studies Hybridoma Bank (DSHB) for providing the NKX6-1 antibody.

## Author contributions

**Conceptualization:** Markus Breunig, Stefan Liebau, Alexander Kleger.

**Data curation:** Markus Breunig, Meike Hohwieler, Eleni Zimmer, Medhanie A. Mulaw, Eric Simon.

**Formal analysis:** Markus Breunig, Meike Hohwieler, Felix von Zweydorf, Natalie Hauff, Cécile Julier, Christian Johannes Gloeckner.

**Funding acquisition:** Markus Breunig, Christian Johannes Gloeckner, Stefan Liebau, Alexander Kleger.

**Investigation:** Markus Breunig, Meike Hohwieler, Jasmin Haderspeck, Felix von Zweydorf, Natalie Hauff, Lino-Pascal Pasquini, Christoph Wiegreffe.

**Project administration:** Markus Breunig, Stefan Liebau, Alexander Kleger.

**Resources:** Christoph Wiegreffe, Medhanie A. Mulaw, Cécile Julier, Eric Simon, Christian Johannes Gloeckner, Stefan Liebau, Alexander Kleger.

**Software:** Markus Breunig, Eleni Zimmer, Medhanie A. Mulaw, Eric Simon.

**Supervision:** Markus Breunig, Meike Hohwieler, Christian Johannes Gloeckner, Stefan Liebau, Alexander Kleger.

**Validation:** Markus Breunig.

**Visualization:** Markus Breunig.

**Writing – original draft:** Markus Breunig.

**Writing – review & editing:** Meike Hohwieler, Jasmin Haderspeck, Lino-Pascal Pasquini, Christoph Wiegreffe, Eleni Zimmer, Cécile Julier, Stefan Liebau, Alexander Kleger.

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
