## [Decision Letter · Decision Letter 0]

11 Sep 2024

Dear Dr Kleger,

Thank you very much for submitting your Research Article entitled 'PPDPF is not a key regulator of human pancreas development' to PLOS Genetics.

The manuscript was fully evaluated at the editorial level and by independent peer reviewers. The reviewers appreciated the attention to an important problem, but raised some substantial concerns about the current manuscript. Based on the reviews, we will not be able to accept this version of the manuscript, but we would be willing to review a much-revised version. We cannot, of course, promise publication at that time.

If you decide to revise the manuscript for further consideration at PLOS Genetics, please aim to resubmit within the next 60 days, unless it will take extra time to address the concerns of the reviewers, in which case we would appreciate an expected resubmission date by email to plosgenetics@plos.org.

If present, accompanying reviewer attachments are included with this email; please notify the journal office if any appear to be missing. They will also be available for download from the link below. You can use this link to log into the system when you are ready to submit a revised version, having first consulted our Submission Checklist .

PLOS has incorporated Similarity Check , powered by iThenticate, into its journal-wide submission system in order to screen submitted content for originality before publication. Each PLOS journal undertakes screening on a proportion of submitted articles. You will be contacted if needed following the screening process.

To resubmit, log into your Editorial Manager account and select the option 'Revise Submission' in the 'Submissions Needing Revision' folder.

We are sorry that we cannot be more positive about your manuscript at this stage. Please do not hesitate to contact us if you have any concerns or questions.

Yours sincerely,

Christine Wells

Academic Editor

PLOS Genetics

Fengwei Yu

Section Editor

PLOS Genetics

Reviewer's Responses to Questions

**Comments to the Authors:**

Reviewer #1: In this manuscript titled ‘PPDPF is not a key regulator of human pancreas development’, the authors seek to evaluate the role of PPDPF in mammalian and human pancreatic development. The premise is the critical nature in zebrafish with sparse data in mammals with the exception of correlation with various tumors.

The authors proceeded through a series of logical experiments to support their assertion. They identified that PPDPF is expressed (via ISH) in mouse fetal pancreas, stomach, hindgut, duodenum and kidney, and that the human proteome map showed broad expression of PPDPF in fetal and adult organs (including pancreas). Interestingly, they identified that PPDPF genomic region was bound by pioneer factors (e.g., FOXA1/2) but not pancreas specific transcription factors. They then used hPSCs and differentiated into pancreatic progenitors along with genetically modified hESC line (PPDPF knock-out, 2 separate clones) for the subsequent analyses.

Using multiple orthogonal approaches (e.g., Flag-tagged / Co-IP assay, RNAseq of WT vs KO clones, human single-cell data, orthotopic implant of cell lines), the investigative team was able to convincingly demonstrate that PPDPF does not carry a substantive role in pancreatic lineage development.

The results are rigorous and clearly laid out, the discussion highlights key aspects of the findings, and the methods are comprehensive. Overall, this manuscript was very nicely prepared. There are only two minor points, and one consideration for the authors (the consideration is not necessary):

Minor: Page 9, paragraph 2, line 8. Were these the hPSCs or the hESC cell line clones? The presumption is that you are referring to the hESC clones and not the hPSCs described in the prior paragraphs (as indicated in figure 3).

Page 16, paragraph 1, second-to-last line: “associated SNPs did not correlate with actual diabetes risk but not with actual diabetes risk” is confusing and perhaps a mis-typed sentence.

**One consideration could be to generate a PPDPF KO mouse model and evaluate pancreatic development in vivo.

Reviewer #2: The manuscript entitled “PPDPF is not a key regulator of human pancreas development” by Breunig et al. focuses on characterising the role of PPDPF in human pancreatic development primarily by use of a stem cell differentiation system.

I recommend the authors to address the following points:

Major:

- The Western blot shown in figure 2D is of poor quality. It is difficult to tell is a PPDPF expression is present in the PP cells, or if is is just background.

- Is the promoter of PPDPF active at earlier stages than PP stage? The histone marks are shown for PP stage (Fig. 2F) but the data in Fig. 2C shows protein expression earlier at the PE stage?

- In figure 3 it is difficult to see that PDX1 staining in the immunofluorescence images, so these appear discordant with the flow cytometry results. Either better quality immunofluorescence images could be used to show PDX1 expression, or consider the use of single channel images.

- The authors mention the upregulation of genes associated with placenta in the RNAseq data associated with Figure 6/Supp Figure 4. Considering that placenta is derived from the trophectoderm, and in Figure 2, the authors show 99.9% of cells are definitive endoderm, can the authors comment on this seeming discordance?

- The transcriptomic changes for NKX6-1 expression are very small (Fig. 6F) but the protein changes (Fig. 2B) are quite large. Can the authors address why they think this is seen?

- In figure 7a, cell counts are performed to assess total cell number. At the PE stage, the PPDPF ko cell lines shown a significant increase in cell number; however, the are no significant changes to either cell death or proliferation (Fig 7b-g). Can the authors comment on this?

- In the human fetal scRNAseq (Fig. 8C/D), very little change is seen in the expression level of PPDPF over time. Do the authors think the small differences between MP, endocrine and acinar cells (for example) is meaningful?

- Does GP2 expression vary between wt and PPDPF ko cells during PP stage or at time of transplant?

- No control/isotype staining for flow cytometry analysis is shown, and as such, the validity of the gating is unable to be assessed. Additionally, there is no mention of how live/dead cell selection occurred. Could these details as well as the control stains please be shown.

- Overall, the authors conclude that PPDPF knockout results in a delay in differentiation rather than a difference in differentiation capacity. If the PP stage cells were further differentiated in vitro to an endocrine cell stage, does the delay continue? At what point (if any) do the knockout cells catchup to the wild-type cells?

Minor:

- GSE accession number for bulk RNA sequencing data generated currently missing

- Software used for image analysis/processing is missing from methods

- Software used for FACS plot generation is missing from methods

**Have all data underlying the figures and results presented in the manuscript been provided?**

Reviewer #1: Yes

Reviewer #2: **No: ** GSE accession for bulk RNAseq currently missing

PLOS authors have the option to publish the peer review history of their article (what does this mean? ). If published, this will include your full peer review and any attached files.

**Do you want your identity to be public for this peer review?** For information about this choice, including consent withdrawal, please see our Privacy Policy .

Reviewer #1: No

Reviewer #2: No

---

## [Decision Letter · Decision Letter 1]

16 Mar 2025

Dear Dr Kleger,

We are pleased to inform you that your manuscript entitled "PPDPF is not a key regulator of human pancreas development" has been editorially accepted for publication in PLOS Genetics. Congratulations!

Yours sincerely,

Fengwei Yu

Section Editor

PLOS Genetics

Fengwei Yu

Section Editor

PLOS Genetics

Aimée Dudley

Editor-in-Chief

PLOS Genetics

Anne Goriely

Editor-in-Chief

PLOS Genetics

Comments from the reviewers (if applicable):

Reviewer's Responses to Questions

**Comments to the Authors:**

Reviewer #1: The authors have done their due diligence in addressing this reviewers concerns.

**Have all data underlying the figures and results presented in the manuscript been provided?**

Reviewer #1: Yes

PLOS authors have the option to publish the peer review history of their article (what does this mean? ). If published, this will include your full peer review and any attached files.

**Do you want your identity to be public for this peer review?** For information about this choice, including consent withdrawal, please see our Privacy Policy .

Reviewer #1: No

**Data Deposition**

http://datadryad.org/submit?journalID=pgenetics&manu=PGENETICS-D-24-00731R1

**Press Queries**

---

## [Editor Report · Acceptance letter]

PGENETICS-D-24-00731R1

PPDPF is not a key regulator of human pancreas development

Dear Dr Kleger,

We are pleased to inform you that your manuscript entitled "PPDPF is not a key regulator of human pancreas development" has been formally accepted for publication in PLOS Genetics! Your manuscript is now with our production department and you will be notified of the publication date in due course.

With kind regards,

Anita Estes

PLOS Genetics

On behalf of:
